



# 1   A synthesis of SNAPO-CO2 ocean total alkalinity and total dissolved
inorganic carbon measurements from 1993 to 2022.
Nicolas Metzl[1], Jonathan Fin[1,2], Claire Lo Monaco[1], Claude Mignon[1], Samir Alliouane[3],
David Antoine[3,4], Guillaume Bourdin[5], Jacqueline Boutin[1], Yann Bozec[6], Pascal Conan[7,8],
Laurent Coppola[3,8], Frédéric Diaz[*], Eric Douville[9], Xavier Durrieu de Madron[10], Jean-Pierre
Gattuso[3,11], Frédéric Gazeau[3], Melek Golbol[8,12], Bruno Lansard[9], Dominique Lefèvre[13],
Nathalie Lefèvre[1], Fabien Lombard[3,14], Férial Louanchi[15], Liliane Merlivat[1], Léa Olivier[1],
Anne Petrenko[13], Sébastien Petton[16], Mireille Pujo-Pay[7], Christophe Rabouille[9], Gilles
Reverdin[1], Céline Ridame[1], Aline Tribollet[1], Vincenzo Vellucci[8,12], Thibaut Wagener[13], Cathy
Wimart-Rousseau[13,17]
[1] Laboratoire LOCEAN/IPSL, Sorbonne Université-CNRS-IRD-MNHN, Paris, 75005, France
[2] OSU Ecce Terra, Sorbonne Université-CNRS, Paris, 75005, France
[3] Sorbonne Université, CNRS, Laboratoire d'Océanographie de Villefranche, LOV, F-06230 Villefranche-sur-
Mer, France
[4] Remote Sensing and Satellite Research Group, School of Earth and Planetary Sciences, Curtin University,
Perth WA 6845, Australia
[5] School of Marine Sciences, University of Maine, Orono, USA
[6] Station Biologique de Roscoff, UMR 7144 – EDYCO-CHIMAR, Roscoff, France
[7] Sorbonne Université, CNRS, Laboratoire d'Océanographie Microbienne, LOMIC, F-66650 Banyuls-sur-Mer,
France
[8] Sorbonne Université, CNRS, OSU Station Marines, STAMAR, Paris, F-75006, France
[9] Laboratoire des Sciences du Climat et de l'Environnement, LSCE/IPSL, UMR 8212 CEA- CNRS-UVSQ,
Université Paris-Saclay, 91191 Gif-sur-Yvette, France
[10] CEFREM, CNRS-Université de Perpignan Via Domitia, 52 Avenue Paul Alduy, 66860 Perpignan, France
[11] Institute for Sustainable Development and International Relations, Sciences Po, 27 rue Saint Guillaume, F-
75007 Paris, France
[12] Sorbonne Université, CNRS, Institut de la Mer de Villefranche, IMEV, Villefranche-sur-Mer, F-06230, France
[13] Aix Marseille Univ, Université de Toulon, CNRS, IRD, MIO, Marseille, France
[14] Research Federation for the study of Global Ocean Systems Ecology and Evolution, FR2022/Tara GOSEE,
75000, Paris, France.
[15] CVRM: Laboratoire de Conservation et de Valorisation des Ressources Marines, Ecole Nationale Supérieure
des Sciences de la Mer et de l'Aménagement du Littoral (ENSSMAL), Station de recherche de Sidi Fredj,
Algeria
[16] Ifremer, Univ Brest, CNRS, IRD, LEMAR, F-29840 Argenton, France
[17] Marine Biogeochemistry, GEOMAR Helmholtz Centre for Ocean Research Kiel, 24105 Kiel, Germany
[*] Passed away 14/3/2021
*Correspondence to*: Nicolas Metzl (nicolas.metzl@locean.ipsl.fr)
**Abstract**. Total alkalinity ($A_T$) and total dissolved inorganic carbon ($C_T$) in the oceans are important properties
to understand the ocean carbon cycle and its link with climate change (ocean carbon sinks and sources) or global
change (ocean acidification). We present a data-base of more than 44 400 $A_T$ and $C_T$ observations in various
ocean regions obtained since 1993 mainly in the frame of French projects. This includes both surface and water
columns data acquired in open oceans, coastal zones and in the Mediterranean Sea and either from time-series or
punctual cruises. Most $A_T$ and $C_T$ data in this synthesis were measured from discrete samples using the same
closed-cell potentiometric titration calibrated with Certified Reference Material, with an overall accuracy of ± 4
μmol kg$^{-1}$ for both $A_T$ and $C_T$. Given the lack of observations in the Indian and Southern Oceans, we added sea

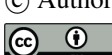



surface underway $A_T$ and $C_T$ data obtained in 1998-2018 in the frame of OISO cruises and in 2019 during the
CLIM-EPARSES cruise measured onboard using the same technique. Separate datasets for the global ocean, and
for the Mediterranean Sea are provided in a single format (https://doi.org/10.17882/95414, Metzl et al., 2023)
that offers a direct use for regional or global purposes, e.g. $A_T$/Salinity relationships, long-term $C_T$ estimates,
constraint and validation of diagnostics $C_T$-$A_T$ reconstructed fields or ocean carbon and coupled climate/carbon
models simulations, as well as data derived from BG-ARGO floats. When associated with other properties, these
data can also be used to calculate pH, fugacity of $CO_2$ ($fCO_2$) and other carbon systems properties to derive
ocean acidification rates or air-sea $CO_2$ fluxes.

**1 Introduction**


Since 1750, humans activities have added 700 (±75) PgC to the atmosphere by burning fossil fuels,

producing cement and changing land use (Friedlingstein et al., 2022) driving up the atmospheric carbon dioxide
($CO_2$) concentration and leading to unequivocal global warming. The ocean plays a major role in reducing the
impact of climate change by absorbing more than 90% of the excess heat in the climate system (Cheng et al.,
2020; von Schuckmann et al, 2020, 2023; IPCC, 2022) and about 25% of human released $CO_2$ (Friedlingstein et
al., 2022). However, the oceanic $CO_2$ uptake changes the chemistry of seawater reducing its buffering capacity
(Revelle and Suess, 1957) and leading to a process known as ocean acidification with potential impacts on
marine organisms (Fabry et al., 2008; Doney et al., 2009, 2020; Gattuso et al., 2015). With atmospheric $CO_2$
concentrations, surface ocean temperature and ocean heat content, sea-level, sea-ice and glaciers, the ocean
acidification (decrease of pH) is now recognized as one of the 7 key properties for global climate indicators
(WMO, 2018). In the frame of the 2030 Agenda, the United Nations established a set of Sustainable
Development Goals (SDG; United Nations, 2020), including a goal dedicated to the ocean (SDG 14, "Life below
water") which calls to "conserve and sustainably use the oceans, seas and marine resources for sustainable
development". Ocean acidification is specifically referred in the SDG indicator 14.3.1 coordinated at the
Intergovernmental Oceanographic Commission (IOC) of UNESCO. Observing the carbonate system in the
oceans and marginal seas and understanding how this system changes over time is thus highly relevant not only
to quantify the global ocean carbon budget, the anthropogenic $CO_2$ inventories or ocean acidification rates, but
also to understand and simulate the processes that govern the complex $CO_2$ cycle in the ocean and to better
predict future climate and global changes (Eyring et al., 2016; Kwiatkowski et al., 2020; Jiang et al., 2023a).

The number and quality of ocean $fCO_2$, $A_T$, $C_T$ and pH measurements has increased substantially over

the past few decades. Quality-controlled observations are now regularly assembled in global data syntheses such
as SOCAT (Surface Ocean $CO_2$ Atlas, Pfeil et al., 2013; Bakker et al., 2014, 2016) and GLODAP (Global Ocean
Data Analysis Project, Key et al., 2004; Olsen et al., 2016, 2019, 2020; Lauvset et al., 2021). These datasets
allow evaluation of properties trends in the global ocean, including the change of the ocean $CO_2$ sink (e.g.
Wanninkhof et al., 2013; Friedlingstein et al., 2022; Watson et al., 2020), anthropogenic $CO_2$ inventories (e.g.
Sabine et al., 2004; Khatiwala et al., 2013; Gruber et al., 2019) and ocean acidification (Lauvset et al., 2015,
2020; Jiang et al., 2019). Thanks to the GLODAP data-base, new methods were recently developed (Sauzède et
al., 2017; Bittig et al., 2018) to reproduce $A_T$ and $C_T$ distributions from other properties like temperature, salinity
and oxygen more often observed in the water column especially from autonomous floats (Claustre et al., 2020;
Mignot et al., 2023). These methods (named CANYON-B and CONTENT, Bittig et al., 2018) are now also used
to help decisions on GLODAP data quality control or to fill in observational gaps (Olsen et al., 2019, 2020;





Tanhua et al., 2019, 2021). The GLODAP data-products were also successfully used to construct new global
ocean $A_T$ and $C_T$ climatological monthly fields in surface and water column using neural network method (e.g.
Broullón et al., 2019, 2020).

Following pioneer works that produced various global-ocean climatologies of the sea-surface carbonate

system (Millero et al., 1998; Lee et al., 2000, 2006; Takahashi et al., 2002, 2009, 2014; Sasse et al., 2013; Jiang
et al., 2019), the coupling of $fCO_2$ data (from SOCAT) and $A_T$ data (from GLODAP) now enables reconstruction
of the full carbonate system in the surface ocean at monthly scale to investigate temporal trends at decadal scale
(e.g. Gregor and Gruber, 2021; Keppler et al., 2023).

International projects such as SOCAT and GLODAP offer important way to synthetize ocean carbon

data. In these projects, each observation is quality controlled offering to users high quality observations for
regional or global analysis, either for processes analysis or to constraint or validate of ocean and coupled
climate/carbon models (CMIP6, e.g. Lerner et al., 2021). SOCAT is a publicly available synthesis product
initiated in 2007 (Metzl et al., 2007) for quality-controlled, surface ocean $fCO_2$ (fugacity of carbon dioxide)
observations made by the international marine carbon research community (Bakker et al., 2016). The first
SOCAT version was released in 2011 (Pfeil et al., 2013; Sabine et al., 2013), followed by 6 SOCAT versions
(Bakker et al., 2014, 2016). The last version in 2023 includes more than 40 million $fCO_2$ data with accuracy
better than 5 µatm (Bakker et al., 2023). One important product from SOCAT is the use of data to estimate
global air-sea $CO_2$ fluxes based on reconstructed $pCO_2$ fields (e.g. SOCOM project, Surface Ocean $pCO_2$
Mapping Intercomparison, Rödenbeck et al., 2015). Since 2015, these results are included each year for the
global carbon budget (Le Quere et al., 2015; Friedlingstein et al., 2022).

On the other hand, following WOCE/JGOFS era in the 90s when almost all observations were started to

be synthetized in a specific recommended format (Joyce and Corry, 1994), GLODAP focusses on water-column
carbon observations (and other properties). Following the original GLODAP data-product (Key et al., 2004), the
project accumulated many new quality controlled observations. One important achievement from GLODAP is
the use of data to estimate the anthropogenic $CO_2$ inventory or its change over decades (Sabine et al., 2004;
Gruber et al., 2019). Both products, SOCAT and GLODAP, are relevant tools to detect oceanic acidification
rates (Lauvset et al., 2015; Jiang et al., 2019).

Although these projects include many international ocean observations there are ocean $CO_2$ related

observations all around the world (published or not published), such as total alkalinity and dissolved inorganic
carbon, not included in SOCAT or GLODAP. This is because SOCAT accepts and controls only $fCO_2$ data,
whereas GLODAP includes and controls water-columns data mainly from WOCE/GO-SHIP/CLIVAR cruises. It
should be noticed that many ocean carbon observations in various formats can be also found in dedicated data-
base such as NCEI/OCADS (former CDIAC-Ocean, Jiang et al., 2013b,
https://www.ncei.noaa.gov/products/ocean-carbon-acidification-data-system), PANGAEA
(https://www.pangaea.de/) or SeaNoe (https://www.seanoe.org/). In this context it is recommended to progress in
data synthesis of the ocean carbon observations that would offer new high quality products for the community
(e.g. for GOA-ON, www.goa-on.org, IOC/SDG 14.1.3, https://oa.iode.org/, Tilbrook et al., 2019).

In this work, we present a synthesis of more than 44 400 $A_T$ and $C_T$ observations obtained over the

1993-2022 period during various cruises or at time-series stations mainly supported by French projects. This
dataset merges observations measured with the same instruments thus being analytically coherent. Most of the
data have accuracy better than ±4 µmol kg$^{-1}$, i.e. between the climate (±2 µmol kg$^{-1}$) and weather (±10 µmol kg$^{-}$





$^{1}$) goals (Bockmon and Dickson, 2015). Hereafter this dataset will be cited as SNAPO-CO2. We describe the
data assemblage and associated quality control and discuss some potential uses of this dataset.

**2 Data collections**

The time series projects and research cruises from which data were collated are listed in Table 1 with
references in the Supplementary file (SNAPO-CO2-cruises) and the sampling locations displayed in Figure 1.
Sampling was performed either from CTD-Rosette casts (Niskin bottles) or from the ship's seawater supply
(intake at about 5m depth depending the ship and swell). Samples collected in 500 mL borosilicate glass bottles
were poisoned with 100 to 300 µL of $HgCl_2$ depending on the cruises, closed with greased stoppers (Apiezon®)
and held tight using elastic band following the SOP protocol (Dickson et al., 2007). Some samples were also
collected in 500 mL bottles closed with screw caps. After completion of each cruise, discrete samples were
returned back to the LOCEAN laboratory (Paris, France) and stored in a dark room at 4 °C before analysis
generally within 2-3 months after sampling (sometimes within a week). Some samples were also measured for
specific processes studies on benthic corals (e.g. Maier et al., 2012; McCulloch et al., 2012) or for mesocosm
and culture experiments but the data are not included in this synthesis as they do not represent natural ocean state
(e.g. addition of Sahara dust during the DUNE project, Ridame et al., 2014).
As opposed to $pCO_2$, surface $A_T$ or $C_T$ observations are generally obtained from discrete sampling
(measured onboard or onshore). Few cruises offer sea-surface semi-continuous $A_T$ or $C_T$ observations (e.g. Metzl
et al., 2006) but new instrumental developments now enable $A_T$ measurements on SOOP lines, Ship of
Opportunity Program (Seelmann et al., 2020). In addition to discrete samples analyzed for various projects
conducted mainly in the North Atlantic, Tropical Atlantic, Tropical Pacific, Mediterranean sea and coastal
regions (Table 1), we complemented this synthesis with $A_T$ and $C_T$ surface observations obtained in the Indian
and Southern oceans during the OISO cruises in 1998-2018 (Metzl et al., 2006; Leseurre et al., 2022; data also
available at NCEI/OCADS: www.nodc.noaa.gov/ocads/oceans/VOS_Program/OISO.html) and the recent CLIM-
EPARSES cruise conducted in the Mozambique Channel in April 2019 (Lo Monaco et al., 2020, 2021). For
OISO cruises the water-column observations are part of the CARINA (CARbon IN the Atlantic) and GLODAP
synthesis products (Lo Monaco et al., 2010; Olsen et al., 2016, 2019, 2020) and not included here. Excepted
when specified, all data in this synthesis were obtained using the same technique used either in laboratory or at
sea (for OISO 1998-2018 and CLIM-EPARSES 2019 cruises).





**Table 1:** List of cruises in the SNAPO-CO2 dataset. This is organized by region: Global Ocean and coastal zones, and Mediterranean Sea (MedSea). See Tables S1, S2, S4 and S4 in the Supplementary Material for a list of laboratories, of CRMs used, for references and for DOI of cruises. Nb = the number of data for each cruise or time-series.

| Cruise/Project | Start | End | Region | Sampling | Nb |
|---|---|---|---|---|---|
| AWIPEV | 2015 | 2021 | Arctic | Surface and sub-surface | 195 |
| SURATLANT+RREX | 1993 | 2017 | North Atlantic | Surface | 2832 |
| OVIDE | 2006 | 2018 | North Atlantic | Surface, Water Column | 397 |
| STRASSE | 2012 | | North Atlantic | Water Column | 205 |
| EUREC4A-OA | 2020 | | North Atlantic | Surface, Water Column | 135 |
| PROTEUS | 2010 | | North Atlantic | Water Column | 27 |
| CHANNEL | 2012 | 2015 | English Channel | Surface | 696 |
| SOMLIT-Brest | 2008 | 2019 | Coastal North Atl | Surface | 1174 |
| SOMLIT-Roscoff | 2009 | 2019 | Coastal North Atl | Surface and 60m | 801 |
| ECOSCOPA | 2017 | 2019 | Coastal North Atl | Surface | 67 |
| PENZE | 2011 | 2020 | River Brittany | Surface and sub-surface | 148 |
| AULNE | 2009 | 2010 | River Brittany | Surface | 27 |
| ELORN | 2009 | 2009 | River Brittany | Surface | 28 |
| BIOZAIRE | 2003 | 2004 | Trop Atlantic | Water Column | 87 |
| EGEE | 2005 | 2007 | Trop Atlantic | Surface | 199 |
| PIRATA-FR | 2009 | 2017 | Trop Atlantic | Surface, Water Column | 513 |
| PLUMAND | 2007 | | Trop Atlantic | Surface | 38 |
| OUTPACE | 2015 | | Trop Pacific | Water Column | 240 |
| PANDORA | 2012 | | Solomon Sea | Water Column | 178 |
| TARA-Pacific | 2016 | 2018 | Trop Pac, NATL | Surface and sub-surface | 325 |
| TARA-Ocean | 2009 | 2012 | Global Ocean | Surface + 400m | 123 |
| TARA-Microbiome | 2021 | 2022 | Atlantic | Surface, Water Column | 216 |
| ACE | 2016 | 2017 | Southern Ocean | Surface, Water Column | 135 |
| MOBYDICK | 2019 | | Southern Ocean | Water Column | 64 |
| CLIM-EPARSES | 2019 | | Indian | Surface | 790 |
| OISO | 1998 | 2018 | South Indian | Surface | 24950 |
| DYFAMED | 1998 | 2017 | MedSea | Water Column | 2118 |
| BOUSSOLE | 2014 | 2019 | MedSea | Surface + 10m | 172 |
| SOMLIT-PointB | 2007 | 2015 | MedSea Coastal | Surface + 50m | 2397 |
| ANTARES | 2010 | 2016 | MedSea | Water Column | 502 |
| MOLA | 2010 | 2013 | MedSea Coastal | Water Column | 66 |
| SOLEMIO | 2016 | 2018 | MedSea Coastal | Water Column | 212 |
| MOOSE-GE | 2010 | 2019 | MedSea | Water Column | 1847 |
| LATEX | 2010 | | MedSea | Water Column | 51 |
| CARBORHONE | 2011 | 2012 | MedSea | Water Column | 706 |
| CASCADE | 2011 | | MedSea | Water Column | 218 |
| DEWEX | 2013 | | MedSea | Water Column | 367 |
| SOMBA | 2014 | 2014 | MedSea | Water Column | 203 |
| AMOR-BFLUX | 2015 | | MedSea Coastal | Water Column | 6 |
| PEACETIME | 2017 | 2017 | MedSea | Water Column | 233 |
| PERLE | 2018 | 2021 | MedSea | Water Column | 805 |

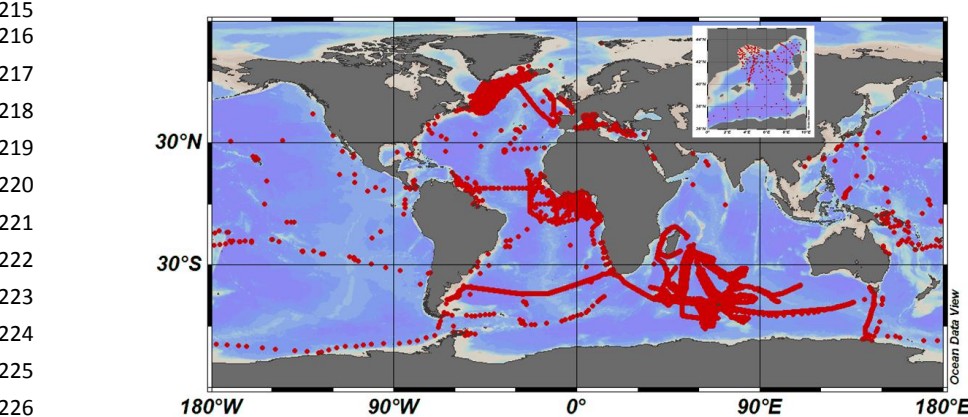

**Figure 1:** Locations of $A_T$-$C_T$ data (1993-2022) in the Global Ocean and the Western Mediterranean Sea (insert). Figure produced with ODV (Schlitzer, 2018).

**3 Method, accuracy, reproducibility, intercomparison and quality control**

**3.1 Method and accuracy**

Except for the DYFAMED time-series observations measured between in 1998 and 2000 in the Mediterranean Sea (Copin-Montégut and Bégovic, 2002; Coppola et al., 2020; Lange et al., 2023), the SURATLANT time-series values acquired from 1993 to 1997 in the North Atlantic subpolar gyre (Reverdin et al., 2018) and samples measured in the river Penzé (Brittany) in 2019-2020 (Yann Bozec, SBR/Roscoff, pers. comm.), all discrete samples were measured at LOCEAN laboratory in Paris (SNAPO-CO2 Service facilities) using the same technique. $A_T$ and $C_T$ were analyzed simultaneously by potentiometric titration using a closed cell (Edmond, 1970; Goyet et al., 1991). In the late 1980s the so-called "JGOFS-IOC Advisory Panel on Ocean CO2" recommended the need for standard analysis protocols and for developing Certified Reference Materials (CRMs) for inorganic carbon measurements (Poisson et al., 1990; UNESCO, 1990, 1991). The CRMs were provided to international laboratories by Pr. A. Dickson (Scripps Institution of Oceanography, San Diego, USA), starting in 1990 for $C_T$ and 1996 for $A_T$, respectively. These CRMs were thus always available to us and used to calibrate the measurements (CRM Batch numbers used for each cruise are listed in Supplementary file (Table S2).

Results of analyses performed on 724 CRM bottles (different Batches) in 2013-2020 are presented in Figure 2. The standard deviations of the differences of measurements were on average around ±3.5 µmol kg$^{-1}$ for both $A_T$ and $C_T$. For unknown reasons, the differences were occasionally up to 10-15 µmol kg$^{-1}$ (0.8% of the data, Figure S2). These few CRM measurements were discarded for the data processing. On average, and excluding some outliers, standard-deviations of the differences for 985 CRM analyses were ±2.95 µmol kg$^{-1}$ for $A_T$ and ±3.30 µmol kg$^{-1}$ for $C_T$, respectively. We did not detect any specific signal for these CRM analyses (e.g., larger uncertainty depending on the Batch number or temporal drifts during analyses, Figure 2) but for some cruises and specific series of samples analyzed over 2 to 7 consecutive days, the accuracy based on CRMs could

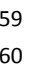

be slightly better than 3 µmol kg$^{-1}$ (±2.5 µmol kg$^{-1}$ for both $A_T$ and $C_T$, e.g. Marrec et al., 2014: Touratier et al.,

2016; Ganachaud et al., 2017; Wimart-Rousseau et al., 2020a).

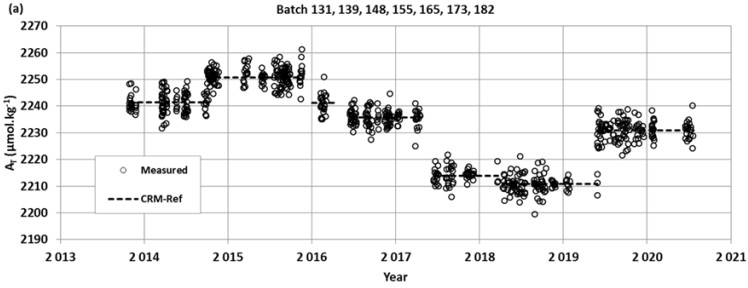

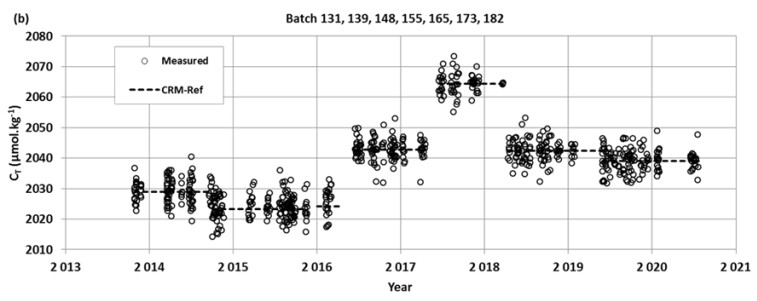

**Figure 2:** $A_T$ (a) and $C_T$ (b) analyses for different CRM Batches measured in 2013-2020. For these 724 analyses
the mean and standard-deviations of the differences with the CRM reference were -0.08 (± 3.35) µmol.kg$^{-1}$ for $A_T$
and 0.15 (± 3.61) µmol.kg$^{-1}$ for $C_T$.

### 3.2 Reproducibility and repeatability

For some projects, duplicates have been regularly sampled (SOMLIT-Point-B, SOMLIT-BREST,
BOUSSOLE/DYFAMED) or replicate bottles sampled at selected depths at fixed stations during the cruises (e.g.
OUTPACE-2015, Wagener et al., 2018; SOMBA-2014, Keraghel et al., 2020). Results of $A_T$ and $C_T$
reproducibility or repeatability are synthetized in Table 2. Figure 3 shows example of regular duplicates from the
times-series SOMLIT-Point-B in the coastal Mediterranean Sea (Kapsenberg et al., 2017), SOMLIT-Brest in the
Bay of Brest, coastal Iroise Sea (Salt et al., 2016) and BOUSSOLE/DYFAMED in the Ligurian Sea (Merlivat et
al., 2018; Golbol et al., 2000, 2020). For the 26 OISO cruises conducted between 1998 and 2018 and the CLIM-
EPARSES cruise in April 2019 (Lo Monaco et al., 2020, 2021), the repeatability was evaluated from duplicate
analyses (within 20 minutes time) of continuous sea surface underway sampling at the same location (when the
ship was stopped). Similarly to what was found for the CRM measurements (Figure S2), differences in
duplicates are occasionally higher than 10-15 µmol kg$^{-1}$ (Figure 3) but most of the duplicates for all projects are
within 0 to 3 µmol kg$^{-1}$. Based on the CRM analyses and replicates for different projects, different regions and
different periods, we estimated the accuracy for both $A_T$ and $C_T$ of ±4 µmol kg$^{-1}$.

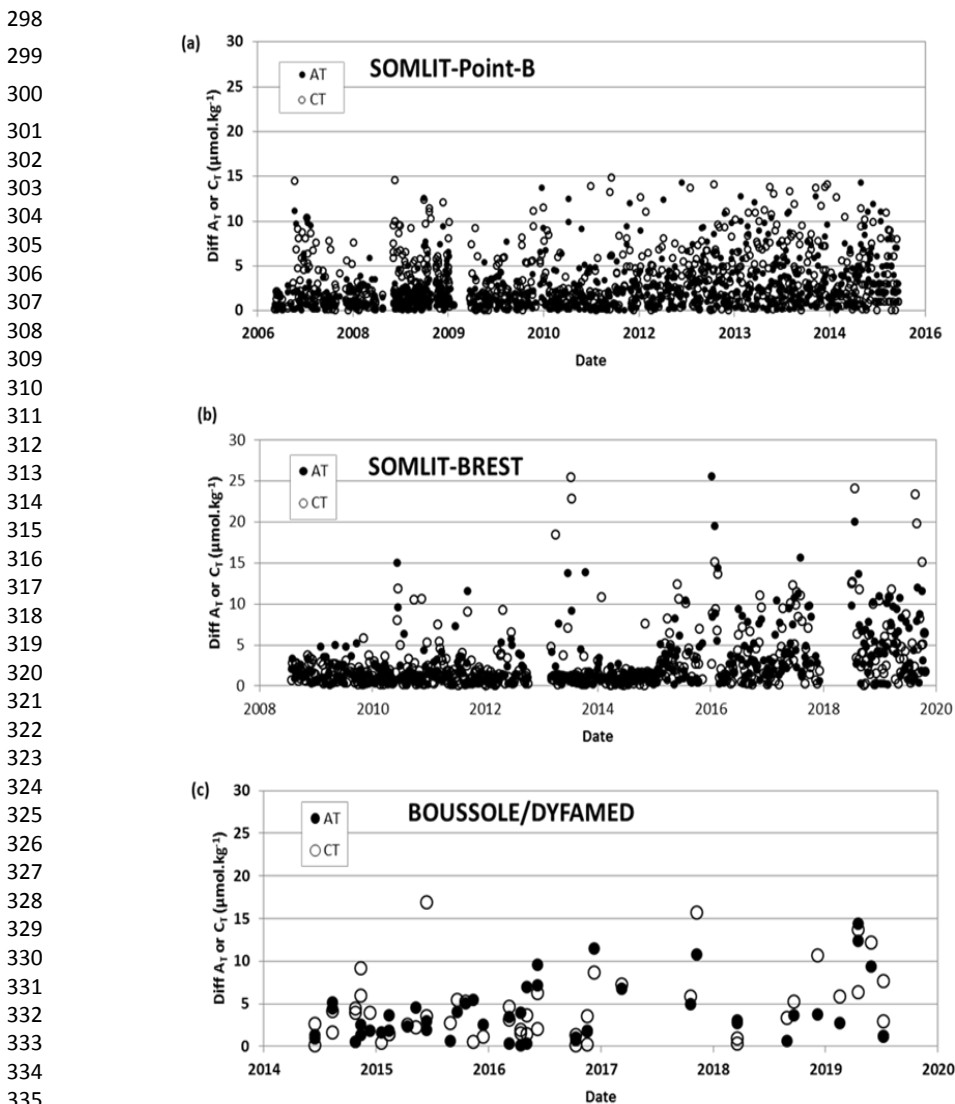

**Figure 3:** Results of duplicate $A_T$ and $C_T$ analyses from the time-series (a) SOMLIT-Point-B in the coastal Mediterranean Sea (Kapsenberg et al., 2017), (b) SOMLIT-BREST in the Bay of Brest, coastal Iroise Sea (Salt et al., 2016 and unpublished) and (c) BOUSSOLE/DYFAMED in the Ligurian Sea (Merlivat et al., 2018; Golbol et al., 2020). The plots show differences in duplicates for both $A_T$ (filled circles) and $C_T$ (open circles). Standard-deviations of these duplicates are listed in Table 2.

**Table 2:** Reproducibility of $A_T$ and $C_T$ analyses for cruises with duplicate analysis. The results are expressed as
the standard-deviations (Std) of the analysis of replicated samples. Nb = the number of replicates for each Time-
series or Cruise. See Figure 4 for the results of regular duplicates for 3 Time-series (SOMLIT-Point-B,
SOMLIT-BREST, and BOUSSOLE). For OISO and CLIM-EPARSES cruises the results correspond to repeated
measurements from continuous sea surface underway sampling at the same location (i.e. within 20 minutes time
and when the ship was stopped). For the 26 OISO cruises (1998-2018) and for simplicity we list the mean
repeatability obtained for all cruises. Detail for each OISO cruise could be consulted in the associated metadata
online at NCEI/OCADS, www.nodc.noaa.gov/ocads/oceans/VOS_Program/OISO.html)

| Project-Cruise | Nb | Std $A_T$ $\mu$mol kg$^{-1}$ | Std $C_T$ $\mu$mol kg$^{-1}$ | Reference |
|---|---|---|---|---|
| OUTPACE | 12 | 3.64 | 3.68 | Wagener et al. (2018) |
| SOMBA | 13 | 2.00 | 3.30 | Keraghel et al. (2020) |
| SOMLIT-Point-B | 786 | 2.63 | 3.10 | Kapsenberg et al. (2017) |
| SOMLIT-Brest | 446 | 3.34 | 3.67 | Salt et al. (2016) + unpub |
| BOUSSOLE | 48 | 3.47 | 4.02 | Merlivat et al. (2018); Golbol et al. (2020) |
| CLIM-EPARSES | 122 | 2.20 | 2.30 | Lo Monaco et al. (2020, 2021) |
| OISO 1998-2018 | 1162 | 2.06 | 2.28 | Metzl et al. (2006) and (*) |

(*) Data available at www.nodc.noaa.gov/ocads/oceans/VOS_Program/OISO.html

**3.3 Inter-comparisons**

Inter-comparisons of measurements performed with different technics help to evaluate the quality of the
data and detect potential shifts (if any) when merging the data in the same region obtained by different
laboratories at different periods. This is especially important to interpret long-term trends of $A_T$ and $C_T$ as well as
for pCO$_2$ and pH calculated with $A_T/C_T$ pairs. For ocean acidification studies, this also refers to the "climate
goal" for which an accuracy for $A_T$ and $C_T$ better than $\pm 2$ $\mu$mol kg$^{-1}$ is needed (Newton et al., 2015; Tilbrook et
al., 2019). Such inter-comparison thus helps to reflect the quality of the data to achieve either the so-called
"weather goal" (for $A_T$ and $C_T$, $\pm 10$ $\mu$mol kg$^{-1}$) or the "climate goal" ($\pm 2$ $\mu$mol kg$^{-1}$) (Bockmon and Dickson,
2015). For the projects in this data synthesis, inter-laboratory comparisons were performed occasionally and
summarized below.
As part of the time-series CHANNEL (2012-2015) in the Western English Channel, Marrec et al.
(2014) analyzed surface samples collected bi-monthly in 2011-2013. $A_T$ analyses were performed with a TA-
ALK-2 system (Appolo SciTech.) while $C_T$ measurements were acquired with an AIRICA system (Marianda
Inc.) Based on CRM analyses (Batch #92) the accuracy was estimated $\pm 3$ $\mu$mol kg$^{-1}$ for $A_T$ and $\pm 1.5$ $\mu$mol kg$^{-1}$
for $C_T$ (Marrec et al., 2014). When comparing with the samples measured at LOCEAN/Paris for the year 2012,
Marrec et al. (2014) concluded that between the two methods the concentrations were within $\pm 2$ $\mu$mol kg$^{-1}$ and
$\pm 3$ $\mu$mol kg$^{-1}$ for $A_T$ and $C_T$ respectively. This is close to the "climate goal" offering confident results for long-
term trend analysis of the carbonate system in this region.
In the frame of the SURATLANT project in the Sub-Polar North Atlantic gyre, some samples collected
at the same time (in 2005, 2006, 2010, 2015, and 2016) were also analyzed onshore for $A_T$ and/or $C_T$ by other
laboratories using different technics (e.g. coulometric method) and the results summarized by Reverdin et al.
(2018). For $C_T$, the mean differences between LOCEAN values and from 4 other laboratories range between -0.7
($\pm 4.6$) and -6.5 ($\pm 3.4$) $\mu$mol kg$^{-1}$ depending on the cruise. For $A_T$ the mean differences with 2 other laboratories



range from -0.6 (±4.1) µmol kg$^{-1}$ to +2.3 (±4.8) µmol kg$^{-1}$. These results range between the "climate goal" and
the "weather goal". See Reverdin et al. (2018) for details on these inter-comparisons.
During OVIDE cruises conducted since 2002 in the North Atlantic along a section from Greenland to
Portugal (Lherminier et al., 2007; Mercier et al., 2015) samples have been taken (since 2006) to complement, for
summer, the SURATLANT time-series in the North Atlantic Sub-Polar gyre (NASPG). The OVIDE samples at
the surface and along the water-column at a few stations were measured back at LOCEAN for $A_T$ and $C_T$ (Metzl
et al., 2018). This enables us to compare our data with the measurements performed on-board by the IIM group
in Vigo/Spain (e.g. Pérez et al., 2010, 2013, 2018; Vazquez-Rodriguez et al., 2012). The OVIDE data have been
regularly quality controlled in CARINA and GLODAP data products (Velo et al., 2009; Key et al., 2010; Olsen
et al., 2016, 2019, 2020). The results of inter-comparisons are gathered in Table 3. For OVIDE in 2006 we
identified (for unknown reason) a large difference between our original $A_T$ values compared to the $A_T$ data
qualified in GLODAP and we thus corrected our $A_T$ data by +7.2 µmol kg$^{-1}$. However, no correction was applied
for $C_T$. For other OVIDE cruises, differences for $A_T$ range between -4.5 (±4.11) µmol kg$^{-1}$ and -0.05 (±3.43)
µmol kg$^{-1}$ depending on the cruise (i.e. $A_T$ measured at LOCEAN was always slightly lower than onboard
measurements). For $C_T$, we compared our measurements onshore with $C_T$ values calculated with $A_T$ and pH
measured onboard. Most of the mean $C_T$ differences are slightly positive (i.e. $C_T$ measured at LOCEAN was
always higher, except for 2010). Taking into account all errors associated with the sampling, the transport of
samples, the instrumentations, the data processing, or the calculations for $C_T$, the comparisons between
LOCEAN and IIM data for OVIDE cruises are deemed acceptable and large differences for both $A_T$ and $C_T$ (> 4
µmol kg$^{-1}$) are far from being systematic (Table 3). The data from SURATLANT and OVIDE can then be
merged to complete the time-series in the NASPG in summer and to better describe the seasonality of the
oceanic carbonate system. For example, in 2010, when the North Atlantic Oscillation (NAO) was strongly
negative, the SURATLANT data showed a rapid decrease of $C_T$ concentrations in the NASPG between early
June and August (Figure 4), with $C_T$ concentrations in August much lower than other years (Racapé et al., 2014).
This leads to a rapid drop in fCO$_2$ in 2009-2010, such that the NASPG was a strong CO$_2$ sink (Leseurre et al.,
2020). The winter-to-summer seasonal decrease of $C_T$ in 2010 in the north NASPG was on average -77 µmol
kg$^{-1}$ (Figure 4) much larger than in the climatology (range -50 to -55 µmol kg$^{-1}$, Takahashi et al., 2014; Reverdin
et al., 2018). The OVIDE data in late June 2010 and SURATLANT in August 2010 confirmed this signal that
was linked to a pronounced primary productivity in that period (Figure 4, Henson et al., 2013; Racapé et al.,
2014; Mc Kinley et al., 2018). Notice that for this period no pCO$_2$ observations were available in July-September
2010 in SOCAT data-product and the $A_T$-$C_T$ data presented here could be used to calculate pCO$_2$ to complement
the pCO2 dataset in this region like was done for other periods (Mc Kinley et al., 2011).

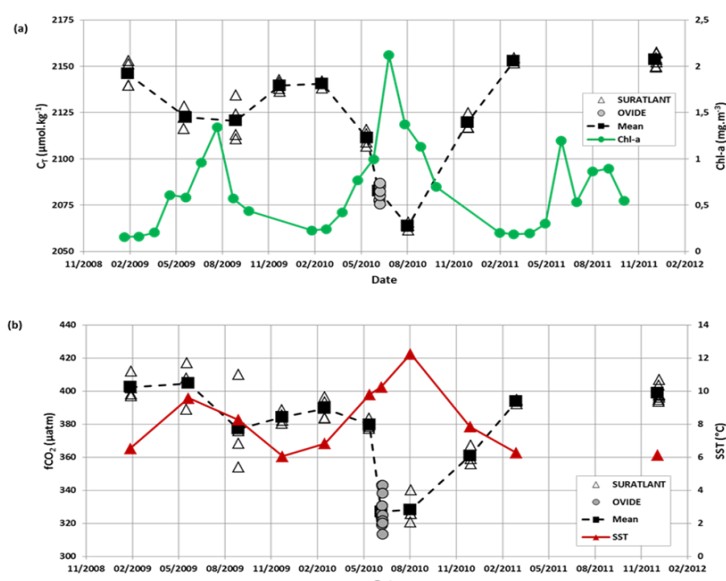

**Figure 4:** (a) Time-series of $C_T$ concentrations (µmol.kg$^{-1}$) for 2009-2011 in surface waters in the North Atlantic Sub-Polar gyre (zone 59°N-33°W) based on SURATLANT (open triangles) and OVIDE-2010 (grey circles) data. In 2009, SURATLANT data were available in February, June, September and December, while in 2010 data available in March, June, August and December and in 2011 data only available for March and December. The OVIDE data in late June 2010 completed the temporal cycle and confirmed the strong seasonal signal and low $C_T$ concentrations in summer 2010 not seen in 2009 (or in 2011 as there is no data in summer). The mean observations for each period describe the $C_T$ seasonal cycles in 2009 and 2010 (Black squares, dashed line). The monthly surface chlorophyll-a concentrations (Chl-a, mg.m$^{-3}$) averaged in the same region based on MODIS are also shown (Green dots and line) highlighting the high productivity during the summer 2010. Chl-a monthly data extracted from MODIS (Giovanni/NASA, last access 3/5/19). (b): Time-series of $fCO_2$ (µatm) for the same cruises (same symbols) calculated with $A_T$-$C_T$. Mean SST (°C) indicated (red triangles). In June 2010 oceanic $fCO_2$ decreased by 53 µatm in 2 weeks.

**Table 3:** Comparisons of $A_T$ and $C_T$ samples measured back at LOCEAN with measurements onboard by IIM Laboratory (F. Pérez, Vigo, Spain) for OVIDE cruises in the North Atlantic. Nb= Number of samples. ND= No Data. The results listed indicate the mean and standard deviations of the differences (LOCEAN-IIM). For $A_T$, IIM values were measured on-board. For $C_T$, IIM values were calculated from $A_T$ and pH both measured onboard. The IIM data were quality controlled and here taken from the GLODAP data-products (Olsen et al, 2016, 2019).

| Cruise Year | Nb $A_T$ | $A_T$ (LOCEAN) – $A_T$ (IIM) µmol kg$^{-1}$ | Nb $C_T$ | $C_T$ (LOCEAN) – $C_T$ (IIM) µmol kg$^{-1}$ |
|---|---|---|---|---|
| OVIDE-2006 | 14 | -2.04 (± 5.84) (*) | 14 | 1.12 (±2.49) |
| OVIDE-2008 | 29 | -4.53 (± 4.11) | 29 | 3.76 (±3.11) |
| OVIDE-2010 | 41 | -1.96 (± 2.26) | 41 | -2.42 (±3.35) |
| OVIDE-2012 | 37 | -0.12 (± 8.85) | ND | ND |
| GEOVIDE-2014 | 57 | -0.05 (± 3.43) | 54 | 2.36 (±7.89) |

(*) for the OVIDE 2006 cruise original difference for $A_T$ was -9.0 (± 5.8) µmol kg$^{-1}$ and LOCEAN $A_T$ data were corrected by +7.2 µmol kg$^{-1}$ based on the mean concentrations in deep layers. No corrections were applied for $A_T$ and $C_T$ for other cruises.

The comparisons described above concern the open ocean region with $A_T$ and $C_T$ concentrations in a

range of concentrations close to the CRM references (used by the different laboratories). Another example of
comparison is presented here for samples obtained along a river and thus for waters with low salinity and $A_T$
concentrations (river Penzé in North Brittany). In 2019, $A_T$ was measured at SBR laboratory (Station Biologique
de Roscoff) by a potentiometric method (using a Titrino-847 plus Metrohm) calibrated with CRM (Batch #131)
for a final accuracy of ±2.1 µmol kg$^{-1}$ (Gac et al., 2020). Although the samples were measured with different
technics the $A_T$/Salinity relationships are very coherent for both datasets (Figure 5). Therefore we added the $A_T$
data measured in 2019-2020 to complete the synthesis for this location (river Penzé).


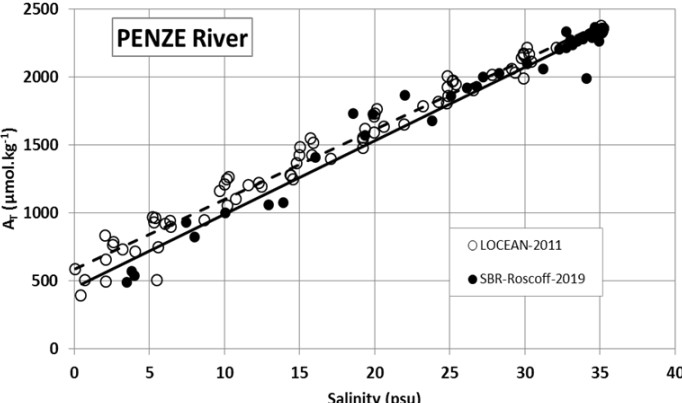

**Figure 5:** Total alkalinity ($A_T$) versus salinity for samples measured in 2011 and 2019 in the river Penzé, North Brittany (Gac et al., 2020). $A_T$ samples were measured at LOCEAN in 2011 (open circles, dashed-line) and at SBR laboratory (Roscoff) in 2019 (filled circles, black line).

### 3.4 Quality control and flags

Identifying each data with an appropriate flag is very convenient for selecting the data (good,

questionable or bad). Here we used 4 Flags for each property (Flags 2, 3, 4, and 9) following the WOCE
program and used in other data products such as SOCAT (Bakker et al., 2016) or GLODAP (Olsen et al., 2016,
2019, 2020; Lauvset et al., 2021). During the data-processing, we first assigned a flag for each $A_T$ and $C_T$ data
based on the standard error in the calculation of $A_T$ and $C_T$ concentrations (non-linear regression, Dickson et al.
2007). By default, if the standard-deviation on the regression is > 1 µmol kg$^{-1}$, we assigned a flag 3
(questionable) although the data could be acceptable and then used for interpretations. Flag 3 was also assigned
when salinity was doubtful or when differences of duplicates were large (e.g. ±20 µmol kg$^{-1}$). Flags 4 (bad or
certainly bad) were assigned when clear anomalies were detected for unknown reason (e.g. a sample probably
not fixed with HgCl$_2$). A secondary quality control was performed by the PIs of each project based on data
inspection, duplicates, $A_T$/Salinity relationship, or the mean observations in deep layers where large variability in
$A_T$ and $C_T$ is unlikely to occur from year to year. An example presents all data from the MOOSE-GE cruises
conducted in 2010-2019 in the Mediterranean Sea (Coppola et al., 2020; Testor et al., 2010) where clear outliers
have been identified (Figure S3). For the 10 MOOSE-GE cruises and a total of 1847 $A_T$ and $C_T$ analyses, 26 were
identified flagged as bad (flag 4), 139 for $A_T$ and 141 for $C_T$ listed as questionable (flag 3) and 1682 for $A_T$ and
1680 for $C_T$ considered as good data (flag 2, i.e. more than 90%). Similar control was performed for each
project.

The synthesis of various cruises in the same region and period also offers verification and secondary
control of the data. For example, several cruises were conducted in the Mediterranean Sea in 2014 (MOOSE-GE,
SOMBA, ANTARES and DYFAMED). The mean values of $C_T$ and $A_T$ in the deep layers (> 1800m) for each
cruise confirmed the coherence of the data (Table 4). This enabled to merge the different datasets for
interpretations of the temporal trends and processes driving the $CO_2$ cycle (Coppola et al., 2019, 2020; Ulses et
al., 2022) or to train and validate a regional neural network to reconstruct the carbonate system (e.g. CANYON-
MED, Fourrier et al., 2020, 2022).
**Table 4:** Mean observations in the deep layers (> 1800m) of the western Mediterranean Sea for different cruises
conducted in 2014. Results in deep layers (> 1800m) for the DEWEX cruise in 2013 and the PEACETIME
cruise in 2017 in the same region are also listed. N-$A_T$ and N-$C_T$ are $A_T$ and $C_T$ normalized at Salinity = 38. Nb =
number of data (with flag 2). Standard-deviations are in brackets. References for these cruises are listed in
Supplementary Material.

| Cruise | Period | Nb | Pot. Temp C | Salinity PSU | N-$A_T$ $\mu$mol kg$^{-1}$ | N-$C_T$ $\mu$mol kg$^{-1}$ |
|---|---|---|---|---|---|---|
| All cruises | Feb/Dec-2014 | 76 | 12.905 (0.007) | 38.486 (0.005) | 2562.9 (5.3) | 2303.7 (4.7) |
| ANTARES | Feb/Nov-2014 | 14 | 12.913 (0.004) | 38.488 (0.006) | 2564.0 (3.8) | 2301.9 (3.5) |
| DYFAMED | Mar/Dec-2014 | 9 | 12.905 (0.0016) | 38.487 (0.004) | 2560.1 (5.0) | 2304.3 (6.8) |
| MOOSE-GE | Jul-2014 | 21 | 12.909 (0.004) | 38.487 (0.005) | 2565.6 (4.6) | 2303.5 (4.1) |
| SOMBA | Aug/Sep-2014 | 32 | 12.899 (0.005) | 38.483 (0.005) | 2561.5 (5.6) | 2304.6 (4.8) |
| DEWEX | Feb/Apr-2013 | 44 | 12.903 (0.010) | 38.588 (0.006) | 2556.0 (4.3) | 2294.0 (5.7) |
| PEACETIME | May/Jun-2017 | 7 | 12.904 (0.002) | 38.486 (0.003) | 2567.2 (10.6) | 2308.1 (8.9) |

The total number of data for the Global Ocean and the Mediterranean Sea are gathered in Table 5 with
corresponding flags for each property. Overall, the synthesis includes more than 94% of good data for both $A_T$
and $C_T$. About 5% are questionable and 2% are likely bad. Overall, we believe that all data (with Flag 2) in this
synthesis have an accuracy better than 4 $\mu$mol kg$^{-1}$ for both $A_T$ and $C_T$, the same as for quality-controlled data in
GLODAP (Olsen et al., 2020; Lauvset et al., 2021). The uncertainty ranges between the "Climate goal" (2 $\mu$mol
kg$^{-1}$) and the "Weather Goal" (10 $\mu$mol kg$^{-1}$) for ocean acidification studies (Newton et al., 2015; Tilbrook et al.,
2019). This accuracy is also relevant to validate or constraint data-based methods that reconstruct $A_T$ and $C_T$



fields with an error of around 10-15 µmol kg$^{-1}$ for both properties (Bittig et al., 2018; Broullón et al., 2019, 2020;
Fourrier et al., 2020: Chau et al., 2023).
**Table 5:** Number of Temperature, Salinity, $A_T$ and $C_T$ data in the synthesis identified for Flags 2, 3, 4, 9. The
data are given for the full data-set Global Ocean and for the Mediterranean Sea. Last column is the percentage of
Flag 2 (Good).

|  | Flag 2 | Flag 3 | Flag 4 | Flag 9 | % Flag 2 |
|---|---|---|---|---|---|
| **Global Ocean** | | | | | |
| Temperature | 43538 | 410 | 0 | 478 | 0.9907 |
| Salinity | 44033 | 319 | 2 | 71 | 0.9928 |
| $A_T$ | 39331 | 2144 | 1165 | 1787 | 0.9224 |
| $C_T$ | 39921 | 2091 | 1148 | 1279 | 0.9250 |
| **Mediterranean Sea** | | | | | |
| Temperature | 9843 | 1 | 0 | 65 | 0.9999 |
| Salinity | 9879 | 8 | 2 | 20 | 0.9999 |
| $A_T$ | 8853 | 425 | 411 | 220 | 0.9137 |
| $C_T$ | 8854 | 451 | 389 | 211 | 0.9133 |

**3.5 Using $A_T$-$C_T$ to calculate pCO$_2$ and pH and compare with pCO$_2$ and pH measurements**

For some projects, the $A_T$-$C_T$ data presented in this synthesis were used to calibrate or validate in situ

pCO$_2$ sensors (Bozec et al., 2011; Marrec et al., 2014; Merlivat et al., 2018). The $A_T$-$C_T$ data were also used to
calculate pCO$_2$ and to derive associated air-sea CO$_2$ fluxes, especially during periods when no direct pCO$_2$
measurements were available (e.g. in the North Atlantic, Figure 4, Watson et al., 2009; Mc Kinley et al., 2011).
For example, Marrec et al. (2014) successfully used the calculated pCO$_2$ (with $A_T$-$C_T$ pairs) to adjust the drift of
the pCO$_2$ data recorded with a Contros-HydroC/CO2 FT sensor mounted on a FerryBox for regularly sampling
the Western English Channel. Here we show the results for the period 2012-2014 (Figure 6). In this region the
alkalinity is relatively constant over time; the average of $A_T$ for 528 samples at different seasons and years is
2334.4 (±7.2) µmol kg$^{-1}$. On the opposite, the $C_T$ concentrations show distinctive seasonality, with higher
concentrations in winter and lower in summer when biological activity is pronounced (Marrec et al., 2013, 2014;
Kitidis et al., 2019). This controls the seasonal pCO$_2$ distribution revealed each year in both measured and
calculated pCO$_2$ (Figure 6). For 528 co-located samples the mean difference between calculated and measured
pCO$_2$ is -1.9 (±11.9) µatm with no distinct differences depending on the season and year.

Earth System
Science
Data

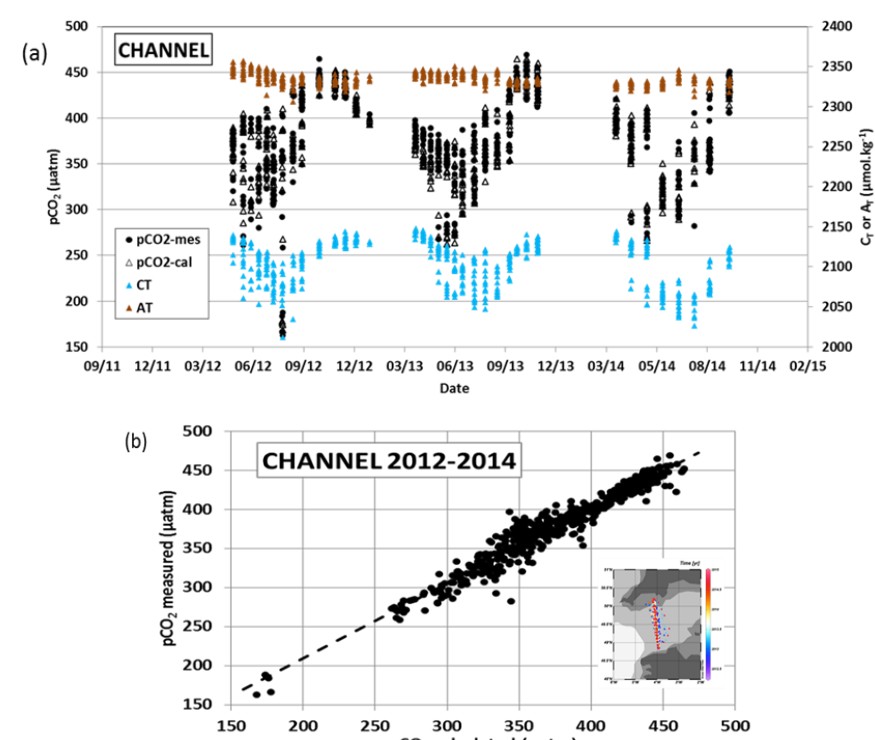

**Figure 6:** (a): Time-series of $A_T$ (brown triangles, right Y-axis), $C_T$ (blue triangles, right Y-axis), $pCO_2$ calculated (open triangles, left Y-axis) and $pCO_2$ measured (filled circles) in the Western English Channel in 2012-2014 (Marrec et al., 2014). (b): Measured $pCO_2$ versus calculated $pCO_2$ for the same samples. The mean difference ($pCO_{2cal}$-$pCO_{2mes}$) for 528 samples is -1.9 µatm (± 11.9) µatm. Data from Marrec and Bozec (2016 a,b; 2017). Localization of the samples is shown in the inserted map.

In the Ligurian Sea, following the first high frequency in situ $fCO_2$ measurements in 1995-1997 at the DYFAMED time-series station (Hood and Merlivat, 2001), a new CARIOCA $fCO_2$ sensor was deployed at that location in 2013 (BOUSSOLE project, Merlivat et al., 2018). The CARIOCA sensor was calibrated with regular $A_T$-$C_T$ analyses performed at LOCEAN. Based on these data, the mean difference between CARIOCA-$fCO_2$ measurements and calculated-$fCO_2$ data was estimated to be around ±4.4 µatm for 2013-2015, i.e. the same order than the precision of the CARIOCA sensor (±5 µatm, Merlivat et al., 2018). Here we extend the results for the period 2013-2018 (Golbol et al., 2020; data also in SOCAT version v2021, Bakker et al., 2016) and compared the CARIOCA $fCO_2$ time-series with $A_T$ and $C_T$ data from different cruises (BOUSSOLE, DYFAMED and MOOSE-GE) selected in the layer 0-20m at that location (Figure S4). For 67 co-located samples at different seasons and years, the mean difference between calculated and measured $fCO_2$ ($fCO_{2cal}$-$fCO_{2mes}$) was -3.7 µatm (± 10.8) µatm. At that location, the alkalinity is relatively constant over 2013-2018 with an average concentration of 2569.8 (±13.2) µmol kg$^{-1}$. $C_T$ concentrations show a clear seasonality, decreasing by around 50 µmol kg$^{-1}$ from winter to late summer driving the large seasonal cycle of $fCO_2$ (range 80 µatm) revealed in both measured and calculated values (here $fCO_2$ is normalized at 13°C, Figure S4). In addition to calibration purposes, a regional $A_T$/Salinity relationship was derived from the $A_T$ data measured at that location and successfully used to construct time-series of $C_T$ and pH calculated from the high-frequency CARIOCA $fCO_2$



data to investigate and interpret the long-term change of $fCO_2$ and acidification in the Ligurian Sea (Merlivat et
al., 2018; Coppola et al., 2020).

672   A CARIOCA sensor was also deployed in 2003 near the SOMLIT-Brest time-series site in the Bay of
Brest (Bozec et al., 2011; Salt et al., 2016). As for BOUSSOLE in the Ligurian Sea, samples collected for $A_T$-$C_T$
were used for validation of the pCO2 recorded by the CARIOCA sensor and the comparison with calculated
$pCO_2$ showed a good agreement, i.e. pCO2cal= 0.98*pCO2mes + 7 µatm (Bozec et al., 2011). CARIOCA
sensors were also deployed on moorings in the Tropical Atlantic (PIRATA project, e.g. Lefèvre et al., 2008,
2016; Parard et al., 2010). With the discrete $A_T$ and $C_T$ data included in this synthesis (EGEE and PIRATA-FR
cruises), the $fCO_2$ data from CARIOCA sensor associated with an adapted $A_T$/Salinity relationship were used to
derive pH (Lefèvre et al., 2016) or $C_T$ time-series to evaluate net community production in the eastern tropical
Atlantic (Parard et al., 2010; Lefèvre and Merlivat 2012).

681   Although this is not a direct instrumental inter-comparison, differences between $pCO_2$ (or $fCO_2$)
calculated using $A_T$-$C_T$ pairs with direct $pCO_2$ measurements give a glimpse of the quality of $A_T$ and $C_T$ data in
this synthesis given the uncertainty attached to the $pCO_2$ or pH calculations (Orr et al., 2015). For example, in
the frame of the SURATLANT project in the North Atlantic, calculated $fCO_2$ data were compared with co-
located $fCO_2$ measurements for different seasons and years (Figure S5). The mean differences ($fCO_{2cal}$-$fCO_{2mes}$)
ranged between -4.3 µatm (± 12.9) µatm (2004-2007, 74 co-located samples) and -3.0 (±12.1) µatm (2014-2015,
98 co-located samples). The differences are almost the same for different years (and seasons) and are thus
attributed to method uncertainties (including sampling time, measurement errors, and data processing). Based on
these comparisons and the consistency between data we are confident that the $A_T$-$C_T$ data presented in this
synthesis could be used to calculate $fCO_2$ (and pH) and interpret temporal changes and drivers of these
parameters as well as to estimate air-sea $CO_2$ fluxes in the North Atlantic (e.g. Corbière et al., 2007; Schuster et
al., 2009, 2013; Watson et al., 2009; Metzl et al., 2010; Mc Kinley et al., 2011; Reverdin et al., 2018, Kitidis et
al., 2019; Leseurre et al., 2020).

694   The $A_T$-$C_T$ data in this synthesis have been also successfully used for $fCO_2$ and air-sea $CO_2$ fluxes
calculations in other regions: the tropical Atlantic (Koffi et al., 2010), the tropical Pacific (Moutin et al., 2018;
Wagener et al., 2018), the Solomon sea (Ganachaud et al., 2017) or the Mediterranean sea and coastal zones (De
Carlo et al., 2013; Marrec et al., 2015; Kapsenberg et al., 2017; Coppola et al., 2020; Keraghel et al., 2020;
Wimart-Rousseau et al., 2020a; Gattuso et al., 2023).

699   In addition, $A_T$-$C_T$ data in the surface and the water-column are also relevant to calculate pH and
evaluate its rate of change for addressing ocean acidification topic in different regions (Kapsenberg et al., 2017;
Ganachaud et al., 2017; Wagener et al., 2018; Coppola et al., 2020; Leseurre et al., 2020; Lo Monaco et al.,
2021). At the time-series station ECOSCOPA in the Bay of Brest (Fleury et al., 2023; Petton et al., 2023), pH
calculated with $A_T$-$C_T$ data were compared with direct pH measurements (Figure S6). In 2017-2019, pH (at
standard temperature 25°C, pH-25C) was always lower than 8 and presented a large seasonal signal of 0.3 (high
pH values in spring, low in winter). The mean difference between calculated and measured pH-25C for 46
samples was equal to +0.013 (± 0.010) which is in the range of the pH uncertainty evaluated by error
propagation when calculated from $A_T$-$C_T$ pairs ($A_T$-$C_T$ error of ±3 µmol kg$^{-1}$ leads to pH error of ±0.0144). Part
of these $A_T$-$C_T$ data used to calculate pH also helped for interpreting the response of marine species to
acidification, e.g. pteropodes or coccolithophores (*Emiliana huxleyi)* in the Mediterranean Sea (Howes et al.,
2015, 2017; Meier et al., 2014) or in the Southern Ocean (Beaufort et al., 2011). The $A_T$-$C_T$ data were also
supporting environmental analysis in coral reef ecosystems in the tropical Pacific (TARA Expedition, Douville
et al., 2022; Lombard et al., 2023; Canesi et al., 2023).

**4 Spatial distribution of $A_T$ and $C_T$: a global view from the SNAPO-CO2 dataset**

The surface distribution in the global ocean based on the SNAPO-CO2 dataset is presented in Figure 7
for $A_T$ and $C_T$. In the open ocean, high $A_T$ concentrations are identified in the subtropics in all basins (Jiang et al.,
2014; Takahashi et al., 2014) with highest concentrations up to 2484 µmol kg$^{-1}$ in the central North Atlantic
(STRASSE cruise in August 2012, 26°N/36°W). In surface and at depth, the $A_T$/Salinity and $A_T$/$C_T$ relationships
are clearly identified and structured at regional scale (Figure 8).





























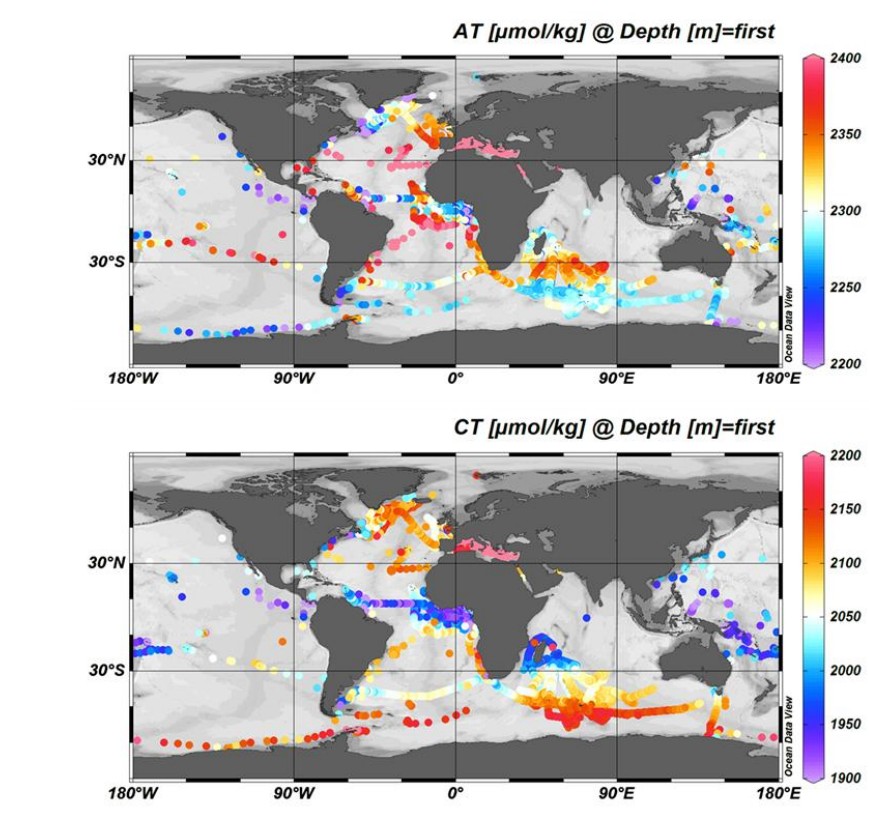

**Figure 7**: Distribution of $A_T$ (top) and $C_T$ (bottom) concentrations (µmol.kg$^{-1}$) in surface waters (0-10m). Only
data with flag 2 are presented in these figures. Figures produced with ODV (Schlitzer, 2018).



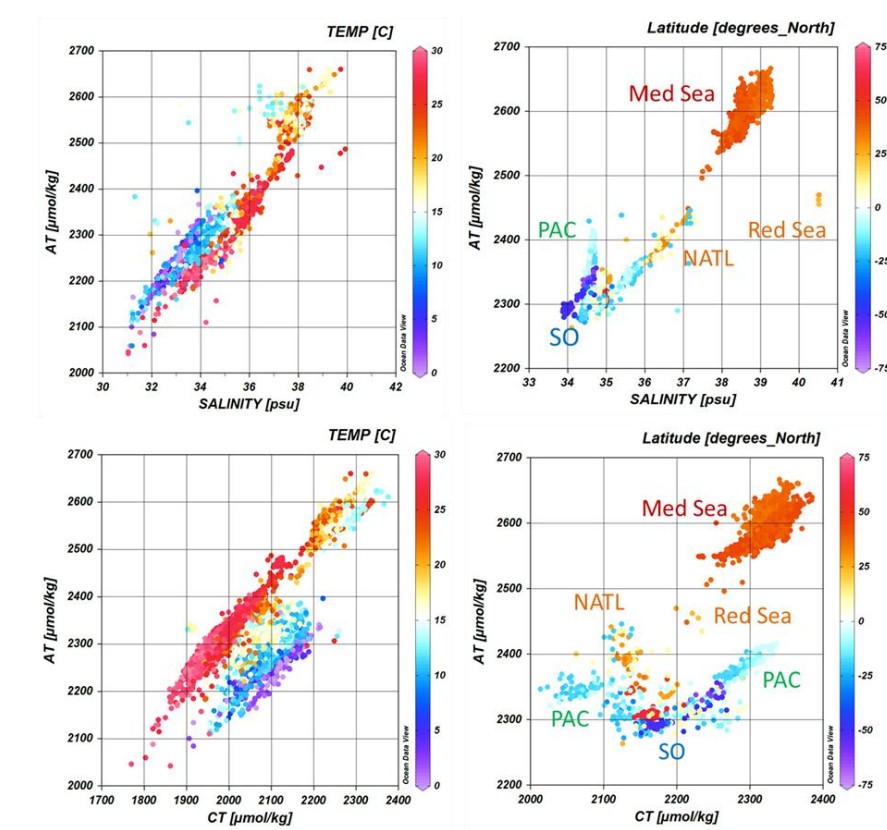

**Figure 8:** Relationships between $A_T$ and Salinity (upper panel) and $A_T$ versus $C_T$ (lower panel) for samples in surface waters (0-10m and SSS > 31) (left) and in the water column below 100m (right). Only data with flag 2 are presented. The color scales correspond to the temperature (left) or the latitude (right). Some location of data are identified: Mediterranean Sea (Med Sea), Red Sea, Tropical Pacific (PAC), North Atlantic (NATL) and Southern Ocean (SO). Figures produced with ODV (Schlitzer, 2018).

In the eastern tropical Atlantic (ETA) where the Congo River impacts the salinity field (Vangriesheim et al., 2009), $A_T$ concentrations range between 2100 and 2400 μmol kg$^{-1}$. The regional $A_T$/Salinity relationship in the ETA based on data from the EGEE cruises in 2005-2007 (Koffi et al., 2010) is robust and validated with more recent measurements from PIRATA-FR cruises in 2010-2019 (Lefevre et al., 2021). The strong $A_T$/Salinity relationship in the ETA was also recognized using data from the TARA-MICROBIOME cruise in May-July 2022 (Figure S7). Low salinity (< 30) and low $A_T$ (1700-2200 μmol kg$^{-1}$) are also observed in the western tropical Atlantic near the Amazon River plume. The $A_T$/Salinity relationships in both river plume regions are very similar (Figure S7).

For $C_T$, the lowest concentrations were observed in the coastal regions of the Tropical Atlantic, on the eastern side in the Gulf of Guinea (BIOZAIRE cruise in 2003, 6°S/11°E, $C_T$=1390 μmol kg$^{-1}$, Vangriesheim et al., 2009) and on the western side in coastal zone off French Guyana (PLUMAND cruise in 2007, 5°N/51°W, $C_T$=1512 μmol kg$^{-1}$, Lefèvre et al., 2010). Such low $C_T$ concentrations were also observed around 5°N/51°W in the Amazon River plume during the recent EUREC4A-OA cruise in 2020 and the TARA-MICROBIOME cruise in 2021 ($C_T$= 1451 μmol kg$^{-1}$) leading to low oceanic fCO$_2$ (< 350 μatm) and a CO$_2$ sink in this region (Olivier et al., 2022).

The high $C_T$ concentrations were mainly observed in the Southern Ocean (OISO and ACE cruises)
south of the Polar Front around 50°S linked to the upwelling of $C_T$-rich deep water (Figure 7, Metzl et al., 2006;
Wu et al., 2019; Chen et al., 2022). This leads to a high $C_T/A_T$ ratio and a high Revelle factor in the Southern
Ocean (Figure 9, Fassbender et al., 2017). The high $C_T$ content and low temperature in the Southern Ocean also
lead to low calcite and aragonite saturation state ($\Omega$) (Takahashi et al., 2014; Jiang et al., 2015) but at present the
surface ocean is not under-saturated with regard to aragonite (Figure 10); however, under-saturation levels ($\Omega$-
Ar<1) were found around 500 m in the Southern Ocean (ACE cruise in 2017, MODYDICK cruise in 2018),
1000 m in the Tropical Pacific (PANDORA 2012 and OUTPACE 2015 cruises) and 2200 m in the North
Atlantic (OVIDE 2012 and 2014 cruises, see also Turk et al., 2017) (Figure 10). Samples at 400 m from the
TARA-Oceans cruise in 2009-2012 also indicated aragonite under-saturation in the Equatorial Atlantic,
Equatorial Pacific, as well as off South America (73°W-34°S, Chile) associated to equatorial or eastern boundary
upwelling systems (Feely et al., 2012; Lauvset et al., 2020).
In surface, $\Omega$−Ar >3 is found in the latitudinal band 45°S-54°N and $\Omega$−Ar<3, below the critical
threshold of $\Omega$−Ar=3.25 that represents a limit for distribution of tropical coral reefs (Hoegh-Guldberg et al.,
2007) is observed at very few locations in the tropics.

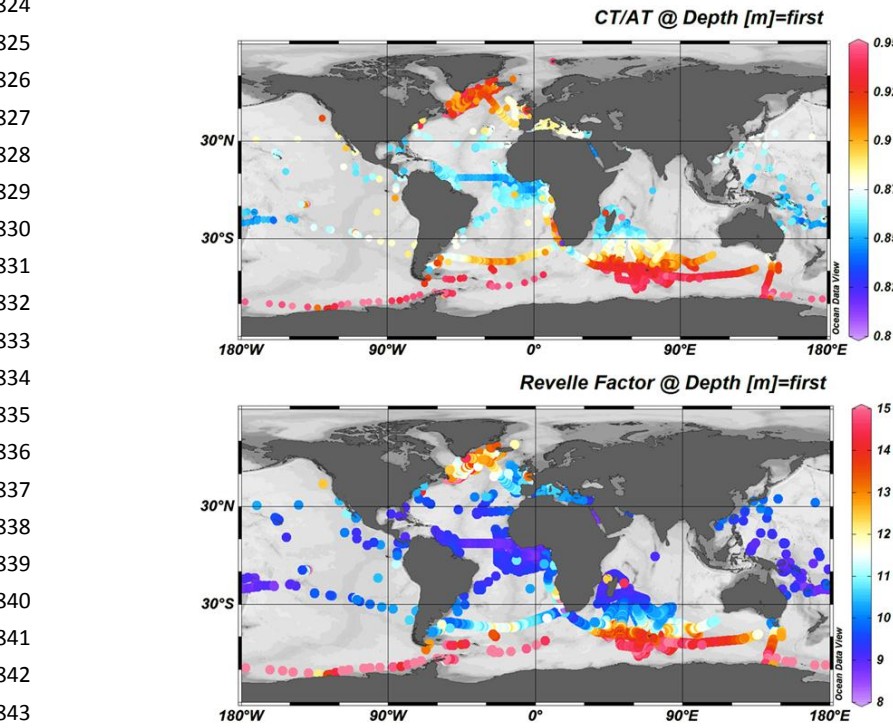

**Figure 9:** Distribution of the $C_T/A_T$ ratio (top) and the Revelle factor (bottom) in surface waters (0-10m). Only data with flag 2 were used. Figures produced with ODV (Schlitzer, 2018).




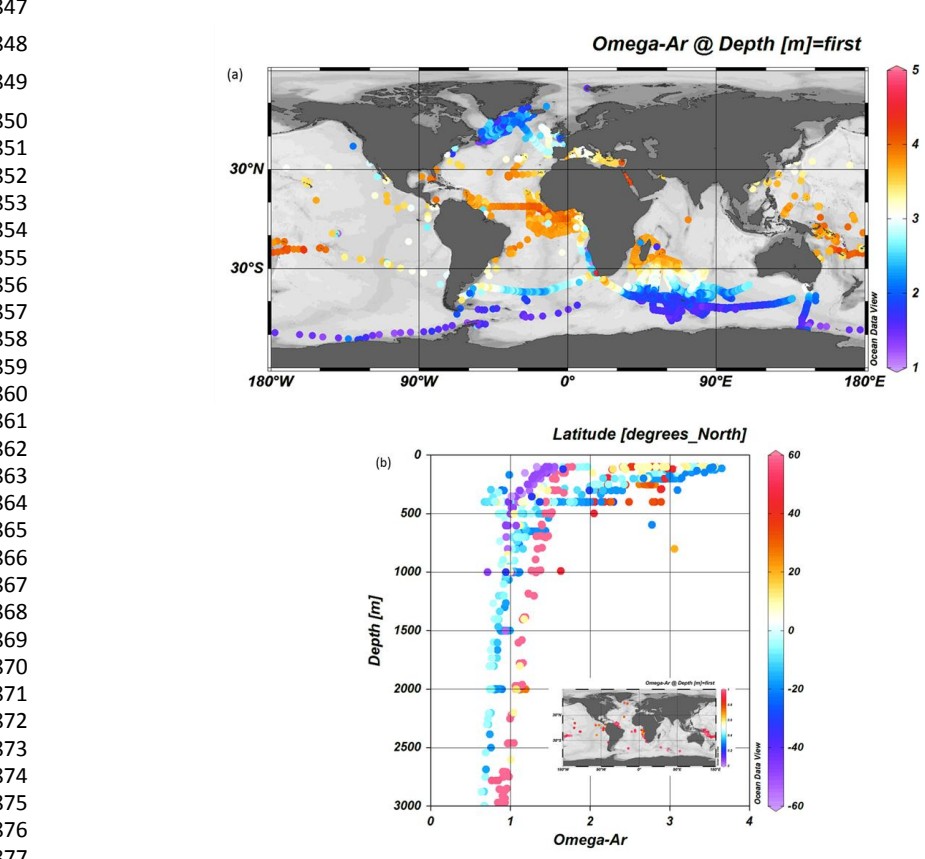

**Figure 10:** (a): Distribution of the aragonite saturation state (Ω-Ar) in surface waters (0-10m). Only data with flag 2 were used. (b): Depth profiles (100-3000m) of Ω-Ar at few locations in the Tropical Pacific, Atlantic and Southern Oceans. Stations where under-saturation is detected (Ω-Ar<1) at depth are identified in the inserted map. Figures produced with ODV (Schlitzer, 2018).

Compared to the open ocean, $A_T$ concentrations are much higher in the Mediterranean Sea (Copin-Montégut, 1993; Schneider et al., 2007; Álvarez et al., 2023) with values up to 2600 µmol kg$^{-1}$ (Figure 8). The $A_T$ and $C_T$ data obtained in 1998-2019 show on average a clear contrast between the northern and southern regions of the western Mediterranean sea (Figure 11 a, b) with higher concentration in the Ligurian Sea and the Gulf of Lion (Gemayel et al., 2015). However, the basin scale average distribution view smoothed the meso-scale signals recognized in the Mediterranean Sea (e.g. Bosse et al., 2017; Petrenko et al., 2017). In the Gulf of Lion the synthesis of 11 cruises conducted from May 2010 to June 2011 (CARBORHONE, CASCADE, LATEX, MOLA, MOOSE-GE) highlights the contrasting distributions of $A_T$ and $C_T$ in the coastal zones and off shore (Figure 11 c, d). The averaging of all data in 1998-2019 also smoothed the seasonal signal and the inter-annual variability described below.


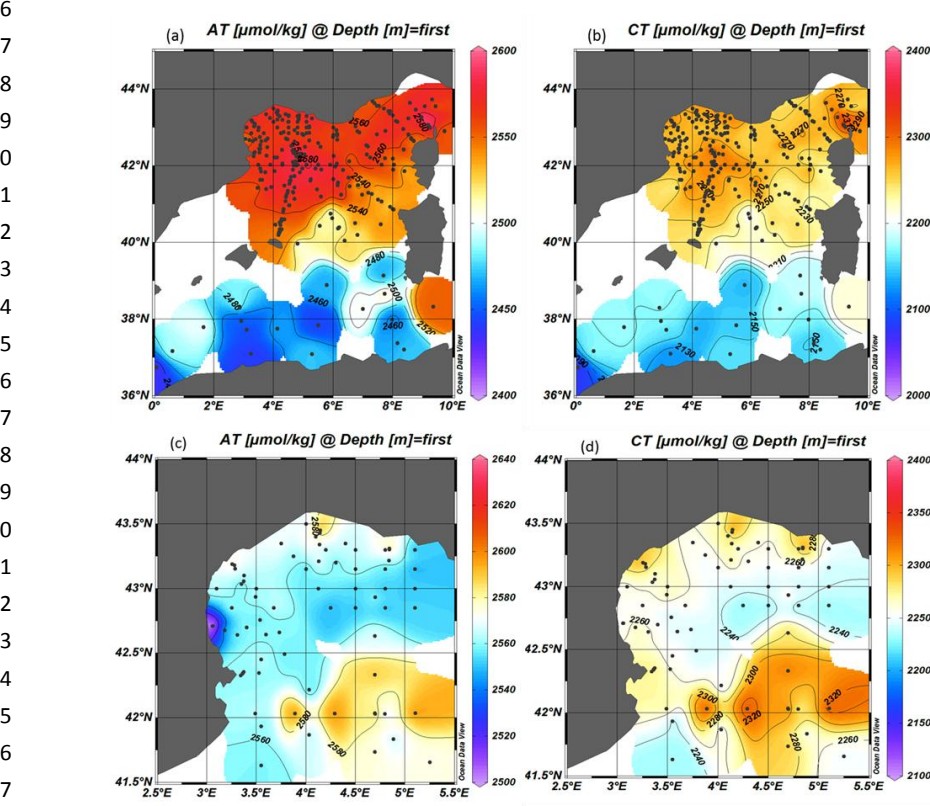

**Figure 11:** Distribution of $A_T$ (a) and $C_T$ (b) in µmol.kg$^{-1}$ in surface waters of the western Mediterranean Sea (0-
10m) from all data for 1998-2019. Detailed distribution of $A_T$ (c) and $C_T$ (d) in µmol.kg$^{-1}$ in surface waters of the
Gulf of Lion for the period 2010-2011 only (cruises CARBORHONE, CASCADE, LATEX, MOLA, MOOSE-
GE). Figures produced with ODV (Schlitzer, 2018).

**5 Temporal variations of $A_T$ and $C_T$:  examples from the SNAPO-CO2 dataset**

Time-series stations such as BATS, ESTOC, HOT, in the Irminger Sea or in the Iceland Sea are the
only way to detect the long-term change in the ocean carbonate system in the surface and the water column
(Bates et al., 2014). These important time-series help to understand driving processes (e.g. Hagens and
Middelburg, 2016) and are often used to validate the pCO$_2$, $A_T$, $C_T$, or pH reconstructed fields (e.g. Rödenbeck et
al., 2013; Broullón et al., 2019, 2020; Keppler et al., 2020; Gregor and Gruber, 2021; Chau et al., 2023; Ma et
al., 2023).
Here we show examples of the temporal surface variations at locations where data were obtained for
more than 10 years (Figure 12). We thus selected the following contrasting regions: the North Atlantic Subpolar
Gyre (NASPG around 60°N/30°°W, period 1993-2018), the Equatorial Atlantic (at 2°N-2°S/12°W-8°W, period
2005-2017), the Indian Ocean subtropical sector (26-35°S/50-56°E, period 1998-2018), the Indian Ocean high
latitude (54-60°S/60-70°E, period 1998-2018), the Ligurian Sea (around DYFAMED station, 43.5-42.5°N/5.5-
9°E, period 1998-2019) and times-series stations in the coastal zones off Britany (period 2008-2019).

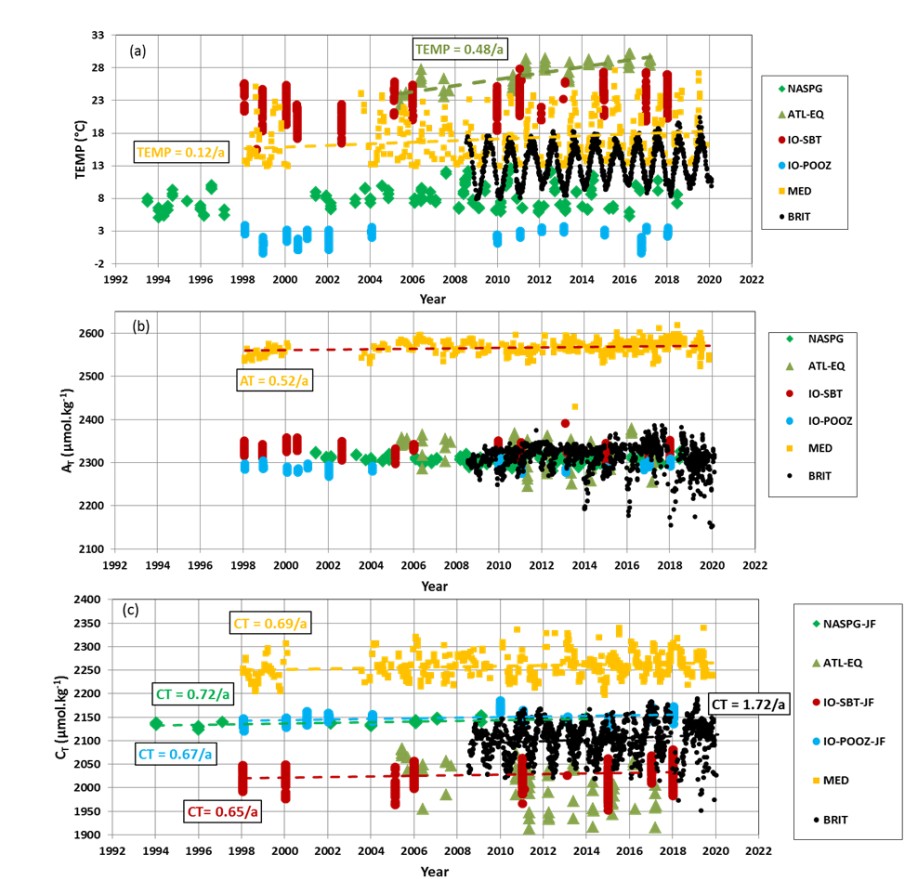

**Figure 12:** Time-series of (a) sea surface temperature (°C), (b) $A_T$ (µmol.kg$^{-1}$) and (c) $C_T$ (µmol.kg$^{-1}$) in 6 regions: the North Atlantic Subpolar Gyre (NASPG 1993-2018, green diamond), the Equatorial Atlantic (ATL-EQ, 2005-2017, green triangle), the Indian subtropical sector (IO-SBT, red circle) and high latitude (IO-POOZ, blue circle) (1998-2018), the Ligurian Sea (MED, 1998-2019, orange square) and times-series stations in the coastal zones off Brittany (BRIT, period 2008-2019, black dots). Trends (dashed lines) are shown when relevant for the discussion ($C_T$ trends listed in Table 6).

**Table 6:** Trend of $C_T$ (µmol kg$^{-1}$ yr$^{-1}$) and corresponding standard error in 5 selected regions where data were available for more than 10 years (data are shown in Figure 12). The projects/cruises for selection of the data in each domain are indicated.

| Region | Period | $C_T$ trend µmol kg$^{-1}$ yr$^{-1}$ | Season | Projects/Cruises |
|---|---|---|---|---|
| NASPG | 1994-2014 | +0.719 (0.168) | Jan-Feb | SURATLANT |
| Indian SBT | 1998-2018 | +0.646 (0.117) | Jan-Feb | OISO |
| Indian High Lat | 1998-2018 | +0.668 (0.042) | Jan-Feb | OISO |
| Ligurian Sea | 1998-2019 | +0.686 (0.181) | All seasons | DYFAMED, BOUSSOLE, MOOSE-GE |
| Coast Brittany | 2008-2019 | +1.720 (0.281) | All seasons | Brest, Roscoff, ECOSCOPA, PENZE |



In the 6 regions, there was a progressive warming most clearly detected in the Mediterranean Sea (e.g.
Nykjaer, 2009). From 1998 to 2019 the warming in the Ligurian Sea was +0.1208°C yr$^{-1}$ ($\pm$0.0227) (Figure 12).
In the equatorial Atlantic, the apparent rapid increase of temperature of +0.48 °C.yr$^{-1}$ ($\pm$0.04) in 2005-2017 from
the selected data indicated a change in water masses and circulation. The colder sea surface in 2005 was
associated with the so-called Atlantic Cold Tongue (ACT) which was one of the most intense ATC since 1982
(Caniaux et al., 2011). The ACT also leads to significant changes in oceanic fCO$_2$ and air-sea CO$_2$ fluxes (Parard
et al., 2010; Koseki et al., 2023) and explained the high $C_T$ concentrations observed in 2005 in this region
(Figure 12, Koffi et al., 2010).
Alkalinity presents rather homogenous concentrations in the NASGP and the south Indian Ocean. Inter-
annual variability of $A_T$ is pronounced in the equatorial Atlantic ranging between 2245 and 2378 µmol kg$^{-1}$. This
is mainly related to salinity as normalized $A_T$ values (N-$A_T$, for salinity= 35) do not show such inter-annual
variability (Mean N-$A_T$ = 2295.7 $\pm$4.6 µmol kg$^{-1}$, n= 67 for 2005-2017, not shown). In the coastal zones off
Brittany, the $A_T$ is also highly variable (Salt et al., 2016; Gac et al., 2021) ranging between 2150 and 2386 µmol
kg$^{-1}$ (Figure 12).
An interesting signal is the progressive increase of $A_T$ in the Mediterranean Sea. The positive $A_T$ trend
of +0.53 ($\pm$0.11) µmol kg$^{-1}$ yr$^{-1}$ (n=538) in 1998-2019 in the region offshore was also observed at the coastal
station SOMLIT-Point-B in 2007-2015 but with a faster increase of +2.08 ($\pm$0.19) µmol kg$^{-1}$ yr$^{-1}$ (Kapsenberg et
al., 2017). Close to the DYFAMED site, at station SOMLIT-Point-B, the $A_T$ trend was not linked to salinity
temporal changes as a positive N-$A_T$ trend was also reported, +0.52 ($\pm$ 0.07) µmol kg$^{-1}$ yr$^{-1}$ (not shown). Based
on data from the PERLE cruises in 2018-2021 a significant increase in $A_T$ was also identified in the Eastern
Mediterranean Sea (Wimart-Rousseau et al., 2021). Along with the increase of $C_T$ and the warming, the $A_T$
increase would impact on the fCO$_2$, air-sea CO$_2$ fluxes and pH temporal changes (Merlivat et al., 2018).
Processes explaining the $A_T$ increase in the Mediterranean Sea are still unexplained and deserve further
investigations (Coppola et al., 2019).
As expected, because of the anthropogenic CO$_2$ uptake the $C_T$ concentrations increased in most regions
(Figure 12, Table 6). This is identified in the Indian Ocean (in the subtropics and the high latitude), in the
Mediterranean Sea, and in coastal waters off Brittany. However, the signal is more complex in the NASPG. As
previously shown the $C_T$ trend in the NASPG depends on seasons and decades (Metzl et al., 2010; Reverdin et
al., 2018; Fröb et al., 2019; Leseurre et al., 2020). Here we selected only the data in January-February from the
SURATLANT cruises leading a $C_T$ trend of +0.719 ($\pm$0.168) µmol kg$^{-1}$ yr$^{-1}$. Compared to the regions further
north the $C_T$ trend in the NASPG is about half the $C_T$ trends of +1.44 ($\pm$0.23) µmol kg$^{-1}$ yr$^{-1}$ observed in the
Iceland Sea (Olafsson et al., 2009) or +1.48 ($\pm$0.22) µmol kg$^{-1}$ yr$^{-1}$ at station M in the Norwegian Sea (Skjelvan
et al., 2022).
In the coastal zones off Brittany, although there are large seasonal and inter-annual variabilities (Gac et
al., 2021), an annual $C_T$ trend of +1.72 ($\pm$0.28) µmol kg$^{-1}$ yr$^{-1}$ is detected over 10 years (2009 to 2019). The same
is observed in the Mediterranean Sea where the $C_T$ offshore trend of +0.69 ($\pm$0.18) µmol kg$^{-1}$ yr$^{-1}$ is low
compared to what was observed in the coastal zone (SOMLIT-Point-B, +2.97 ($\pm$0.20) µmol kg$^{-1}$ yr$^{-1}$, Kapsenberg
et al., 2017).
In the southern Indian Ocean, $C_T$ concentrations also increased in both subtropical and high latitudes,
two regions where the primary productivity is relatively low (oligotrophic regime in the subtropics and High
Nutrient Low Chlorophyll regime, HNLC, south of the Polar Front). With the data selected for austral summer



(January-February) the $C_T$ trends appeared almost similar in these two regions, around +0.65 µmol kg$^{-1}$ yr$^{-1}$
(Table 6).
Finally, in the Equatorial Atlantic the selected data around 0°-10°W highlighted the large variability
linked to the oceanic circulation. Detecting a $C_T$ trend as well as a possible link with anthropogenic carbon
uptake, at least with the data available in 2005-2017, appears to be intricate as it has been previously discussed
for the period 2006-2013 (Lefèvre et al., 2016). However, the signal of the $C_T$ increase is better identified north
or south of the Equator in the eastern tropical Atlantic sector (Lefèvre et al., 2021).
In the water column $A_T$-$C_T$ data from dedicated cruises were used to evaluate the $C_{ant}$ distribution and
pH change since pre-industrial era (e.g. PANDORA cruise, Ganachaud et al., 2017; OUTPACE cruise, Wagener
et al., 2018; SOMBA cruise, Keraghel et al., 2020). Time-series at DYFAMED station also enabled to
investigate the temporal variability of $C_T$, $A_T$ and $C_{ant}$ in the water column (Touratier and Goyet, 2009; Coppola
et al., 2020; Fourrier et al., 2022). As an example of the observed temporal variations at depth we selected the
data in the layer 950-1050m in the Ligurian Sea from different cruises (Figure 13). At that depth both $A_T$ and $C_T$
present some large anomalies especially noticed in 2013 (lower $A_T$ and $C_T$ in February 2013, DEWEX cruise)
and in 2018 (lower $A_T$ and $C_T$ in May 2018, MOOSE-GE cruise) the later probably linked to episodic convective
process that occurred in winter 2018 (Fourrier et al., 2022). In this region the long-term increase of $A_T$ indicates
that in addition to the anthropogenic $CO_2$ signal another process is at play to explain the rapid $C_T$ trend of +1.20
(±0.12) µmol kg$^{-1}$ yr$^{-1}$ at depth compared to that observed in surface (Figure 12). The signal at depth is probably
linked to the variations of the deep convection and mixing with Levantine intermediate water (LIW, Margirier et
al., 2020) with higher $A_T$ and $C_T$ concentrations. The long-term increase of $A_T$ and $C_T$ at depth (here at 1000m,
Figure 13) was also observed below 2000m (Coppola et al., 2020) a signal that has to be investigated in
dedicated analysis using other properties (O$_2$, nutrients, following Fourrier et al. (2022) for the period 2012-
2020) and a larger dataset in the Mediterranean Sea (e.g. CARIMED).

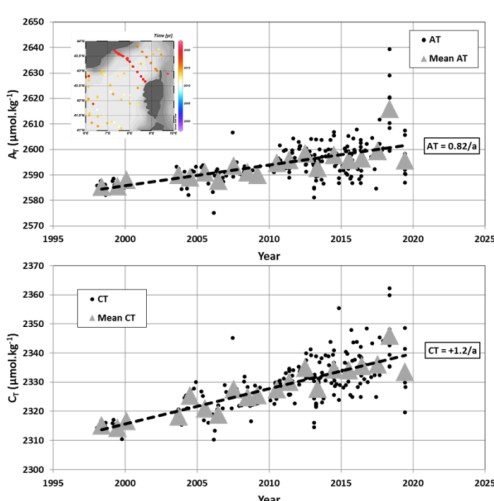

**Figure 13:** Time-series of $A_T$ (µmol.kg$^{-1}$) and $C_T$ (µmol.kg$^{-1}$) in the Ligurian Sea (1998-2019) in the layer 950-1050m. Annual mean (grey triangles) was calculated from all data each year (black dots). The trends (dashed line) based on annual mean are +0.82 (±0.15) µmol.kg$^{-1}$.yr$^{-1}$ for $A_T$ and +1.20 (±0.12) µmol.kg$^{-1}$.yr$^{-1}$ for $C_T$. In this layer data selected are from cruises ANTARES, CASCADE, DEWEX, DYFAMED, MOOSE-GE and PEACETIME (location of stations shown in the inserted map).





**6 Using $A_T$ and $C_T$ data to validate observations from autonomous instruments**

The dataset presented in this synthesis would also offer interesting observations to validate properties ($A_T$ and $C_T$) derived from BG-ARGO floats equipped with pH sensors (e.g. Bushinsky et al., 2019; Mazloff et al., 2023; Mignot et al., 2023). The water column in situ $A_T$-$C_T$ data obtained during the Antarctic Circumpolar Expedition (ACE) in 2016-2017 were collected at location where SOCCOM floats were launched (Walton and Thomas, 2018). A SOCCOM float (WMO ID 5905069) was launched on January 11[th] 2017 at 55°S-96°E south of the Polar Front in the southern Indian Ocean. The pH, temperature and salinity data from the float were then used to derive $A_T$ and $C_T$ profiles (here using the MLR method, Williams et al., 2016, 2017). In the top layers the discrete ACE data (Figure 14) present large variability of $A_T$ and $C_T$ concentrations not captured in the records derived from the float (MLR method somehow smooth the profiles). However, given the uncertainty in reconstructed $A_T$ from float data (5.6 µmol kg$^{-1}$) the average values in the first 100m were almost identical ($A_T$-$_{ACE}$ = 2285.1 (±4.4) µmol kg$^{-1}$ and $A_T$-$_{float}$ = 2278.3 (±0.7) µmol kg$^{-1}$; $C_T$-$_{ACE}$ = 2139.7 (±9.2) µmol kg$^{-1}$ and $C_T$-$_{float}$ = 2141.1 (±3.2) µmol kg$^{-1}$). Moreover below 200m, profiles from the float are coherent compared to the $A_T$-$C_T$ measurements (Figure 14). This is encouraging for using float data to explore the seasonal variability of $A_T$ and $C_T$ in the Southern Ocean (e.g. Williams et al., 2018; Johnson et al., 2022) and the estimation of anthropogenic $CO_2$ in the water column in this sector (Figure 14). Here the $C_{ant}$ concentrations were calculated below 200m (corresponding to the temperature minimum of the winter winter in the SO and using the TrOCA method, Touratier et al., 2007). The float data suggest that $C_{ant}$ concentrations are positive down to about 1000m, with maximum values in subsurface. In 2017 the mean $C_{ant}$ concentration at 200m was 49.1 (±9.0) µmol kg$^{-1}$. Below that depth, $C_{ant}$ decreased and reduced to +29.8 (±8.5) µmol kg$^{-1}$ in the layer 300-400m. To complement the $C_{ant}$ inventories based on GLODAP data-product (e.g. Gruber et al., 2019) $C_{ant}$ estimates derived from BG-ARGO floats as evaluated here in the Southern Ocean could be applied in other locations as was previously tested in the North Pacific (Li et al., 2019).

In surface water as the $A_T$ derived from the float data are deduced using MLR or LIAR methods (Williams et al., 2017; Carter et al., 2016), the $A_T$ data in the SNAPO-CO2 synthesis could also be used to identify $A_T$ anomalies not always captured from floats. This is particularly relevant in coccolithophores blooms areas when low $A_T$ concentrations and high pCO2 are observed (e.g. Balch et al., 2016 in the Southern Ocean; Robertson et al., 1994 in the North Atlantic).

**7 Summary and suggestions**

The ocean data synthesized in this product are based on measurements of $A_T$ and $C_T$ performed in 1993-2022 with an accuracy of ±4 µmol kg$^{-1}$. It offers a large data set of $A_T$ and $C_T$ for the global ocean and regional biogeochemical studies. It includes more than 44 400 surface and water column observations in all oceanic basins, in the Mediterranean Sea, in the coastal zones, near coral reef, and in rivers. For the open ocean this complements the SOCAT and GLODAP data-products (Bakker et al., 2016; Lauvset et al., 2021) and for the Mediterranean Sea the ongoing CARIMED dataset. For the coastal sites this also complements the synthesis of coastal time-series only done around North America (Fassbender et al., 2018; Jiang et al., 2020; OCADS, 2023).



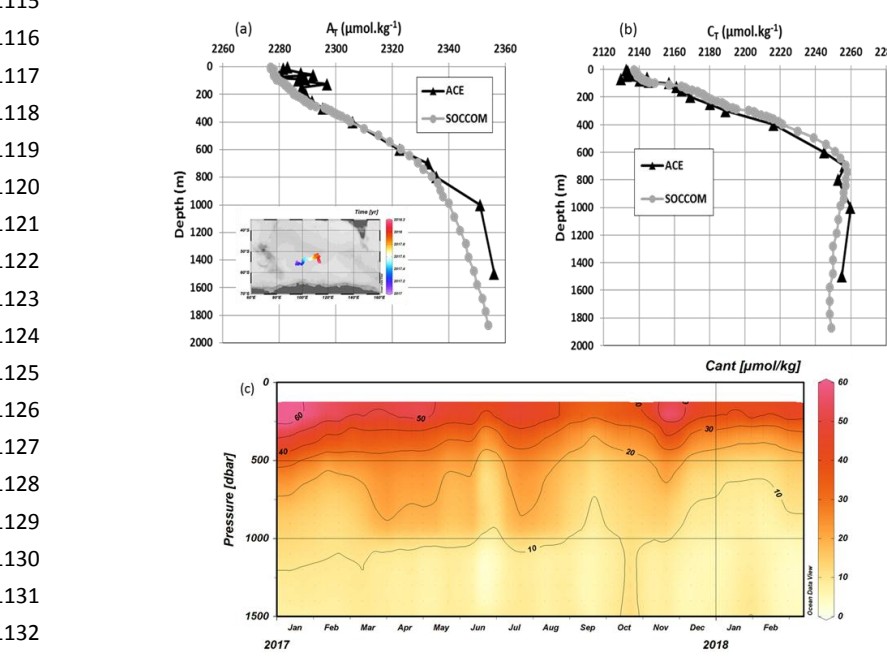

**Figure 14:** Profiles of (a) $A_T$ (µmol.kg$^{-1}$) and (b) $C_T$ (µmol.kg$^{-1}$) observed at station ACE-20 (55°S-95°E, 11/1/17, black triangles) compared with the profiles deduced from the SOCCOM float (WMO code 5905069) launched at that location (first data on January 12$^{th}$ 2017, grey circles). The location/drift of the float in 2017-2018 is shown on the inserted map. (c) Hovmoller section (Pressure/time) of anthropogenic $CO_2$ concentrations ($C_{ant}$ in µmol.kg$^{-1}$) estimated from the float data ($A_T$, $C_T$, O2, T) below 200m (period January 2017-February 2018). Section produced with ODV (Schlitzer, 2018).

The SNAPO-CO2 dataset enables to investigate seasonal variations to decadal trends of $A_T$ and $C_T$ in various oceanic provinces. In regions where data are available for more than 2 decades in surface water (North Atlantic, Ligurian Sea, Southern Indian Ocean, and coastal regions), all time-series show an increase in $C_T$. Excepted in the Mediterranean Sea, $A_T$ appears relatively constant over time, although the $A_T$ content present significant inter-annual variability such as in the NASPG or in the coastal zones including near the Congo and Amazon Rivers plumes.

This dataset represents independent data for validation of reconstructed $A_T$ or $C_T$ fields using various methods (e.g. Rödenbeck et al., 2013, 2015; Sauzède et al., 2017; Turk et al., 2017; Bittig et al., 2018; Broullón et al., 2019, 2020; Land et al., 2019; Keppler et al., 2020; Fourier et al., 2020; Gregor and Gruber, 2021; Sims et al., 2023; Chau et al., 2023). It is also useful to validate Earth System Models (ESM) that currently present bias to reproduce the seasonal cycle of $C_T$ and $A_T$ due to inadequate representation of biogeochemical cycles, including the coupling of biological and physical processes (e.g. Pilcher et al., 2015; Mongwe et al., 2018; Lerner et al., 2021). This should be resolved for confident in future projections of the productivity, ocean acidification, and the responses of the marine ecosystems (e.g. Kwiatowkki et al., 2020). Recall that OBGM or ESM models calculate pCO2 from $A_T$-$C_T$ pairs and the simulated annual $CO_2$ flux might be correct when compared to observations but for wrong reasons (e.g. Goris et al., 2018, Lerner et al., 2021). For example, it has been shown that biases in $A_T$ in ESM models led to an overestimation of the oceanic fCO2 trend and thus uncertainty when predicting the oceanic anthropogenic $CO_2$ uptake (Lebehot et al., 2019). The simulated



seasonal cycle of $pCO_2$ is also uncertain in ESM models especially in high latitudes (e.g. Joos et al., 2023). It is
thus important to attempt validating ESM models with $A_T$-$C_T$ data such as presented in this synthesis.
This dataset would also serve for validating autonomous platforms capable of measuring pH and $pCO_2$
variables and, along with SOCAT and GLODAP datasets, provides an additional reference dataset for the
development and validation of regional biogeochemical models for simulating air-sea $CO_2$ fluxes. It is also
essential for training and validating neural networks capable of predicting variables in the carbonate system,
thereby enhancing observations of marine $CO_2$ at different spatial and temporal scales.
The data presented here are available online on the Seanoe servor (Metzl et al., 2023,
https://doi.org/10.17882/95414) and is divided in two files: one for the Global Ocean, and one for the
Mediterranean Sea. The sources of the original datasets (doi) with the associated references are listed in the
Supplementary Material (Table S3, S4). We invite the users to comment on any anomaly that would have not
been detected or to suggest potential misqualification of data in the present product (e.g. data probably good
although assigned with Flag 3, probably wrong). The SNAPO-CO2 dataset will be regularly updated on Seanoe
data servor with new observations controlled and archived.
**8 Data availability**
Data presented in this study are available at Seanoe: https://www.seanoe.org, https://doi.org/10.17882/95414
(Metzl et al., 2023).

*Author contributions*. NM prepared the data synthesis, the figures and wrote the draft of the manuscript with
contributions from all authors. JF measured the discrete samples since 2014, with the help from CM and CLM,
and prepared the individual reports for each project. NM and JF pre-qualified the discrete $A_T$-$C_T$ data. CLM and
NM are co-Is of the ongoing OISO project and qualified the underway $A_T$-$C_T$ data from OISO cruises. All
authors have contributed either to organizing cruises, sample collection or data qualification.
*Competing interest*. The authors declare that they have no conflict of interest.
*Acknowledgments*. The $A_T$ and $C_T$ data presented in this study were measured at the SNAPO-CO2 facility
(Service National d'Analyse des Paramètres Océaniques du CO2) housed by the LOCEAN laboratory and part of
the OSU ECCE Terra at Sorbonne University and INSU/CNRS analytical services. Support by INSU/CNRS, by
OSU ECCE Terra and by LOCEAN, is gratefully acknowledged as well as support by different French "Services
nationaux d'Observations", such as OISO/CARAUS, SOMLIT, PIRATA, SSS and MOOSE. We thank the
research infrastructure ICOS (Integrated Carbon Observation System) France for funding a large part of the
analyses. We acknowledge the MOOSE program (Mediterranean Ocean Observing System for the Environment,
https://campagnes.flotteoceanographique.fr/series/235/fr/) coordinated by CNRS-INSU and the Research
Infrastructure ILICO (CNRS-IFREMER). AWIPEV-CO2 was supported by the Coastal Observing System for
Northern and Arctic Seas (COSYNA), the two Helmholtz large-scale infrastructure projects ACROSS and
MOSES, the French Polar Institute (IPEV) as well as the European Union's Horizon 2020 research and
innovation projects Jericho-Next (No 871153 and 951799), INTAROS (No 727890) and FACE-IT (No 869154).
Support from the French *Agence Nationale de la Recherche* (ANR) is also acknowledged through their funding
of the BIOCAREX project. The EURECA4-OA cruise was also supported by the EUREC4A-OA JPI Ocean and
Climate program. The OISO program was supported by the French institutes INSU (Institut National des



Sciences de l'Univers) and IPEV (Institut Polaire Paul-Emile Victor), OSU Ecce-Terra (at Sorbonne Université),
and the French program SOERE/Great-Gases. We thank the French oceanographic fleet ("Flotte
océanographique française") for financial and logistic support for all cruises listed in this synthesis and for the
OISO program (https://campagnes.flotteoceanographique.fr/series/228/). Data from the float launched during the
ACE cruise were made freely available by the Southern Ocean Carbon and Climate Observations and Modeling
(SOCCOM) Project funded by the National Science Foundation, Division of Polar Programs (NSF PLR -
1425989), supplemented by NASA, and by the International Argo Program and the NOAA programs that
contribute to it. The Argo Program is part of the Global Ocean Observing System
(http://doi.org/10.17882/42182, http://argo.jcompos.org). We thank Frédéric Merceur (IFREMER) for preparing
the page and data availability on Seanoe. We thank Patrick Raimbault (retired, former at MIO, Marseille) for
managing the MOOSE project until 2019. We thank all colleagues and students who participated to the cruises
and have carefully collected the precious seawater samples. We warmly acknowledge our colleague Christian
Brunet (retired) for his supportive help for the analysis since the start of the Service facility SNAPO-CO2. We
would like to pay tribute to our late colleague Frédéric Diaz who contributed to the LATEX cruise in 2010.

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
