# Peer review of "A synthesis of SNAPO CO2 ocean total alkalinity and total dissolved"

_Earth System Science Data, 2023_

## Referee Comment (RC2)

[referee-annotated manuscript omitted]

---

## Author Comment (AC1)

Response to Reviewers' comments on the manuscript:

A synthesis of SNAPO-CO2 ocean total alkalinity and total dissolved inorganic carbon measurements from 1993 to 2022, Nicolas Metzl et al., MS No.: essd-2023-308, MS type: Data description paper

Reply to Reviewer 1 (in purple from reviewer, in black our reply)

Review 1: posted 15/9/23
The paper by Metzl et al. presents an impressive data synthesis product, encompassing measurements of dissolved inorganic carbon, total alkalinity, and other hydrographic variables collected from both open ocean and coastal regions, as well as the Mediterranean Sea. These data were assembled within the context of various French research initiatives. Overall, the paper is very well written and the data product will contribute significantly to enrich the global ocean carbon observational database. It should be published in a timely manner after incorporating some changes as laid out below.

Reply: We thank the reviewer for her/his enthusiastic support.

Major comments:

1. I recommend dividing the paper into two distinct publications. The first should focus solely on detailing the data product, omitting Figures 8 through 14 and their associated discussions. The content suggested for removal would be more fittingly explored in a separate, specialized paper. Alternatively, they can be put into the supplementary.

Reply: We are not sure to understand the suggestions for two publications (not suggested by reviewer 2). The aim of this manuscript is to describe the data assemblage, the data quality control and to discuss some potential uses of this dataset. We would like to keep the structure of the paper with Figures 8 to 14 as examples for different analysis based on this new data set.

Other publications with specific topics (e.g. detail of seasonality and long-term trends at regional scale, analysis of meso-scale distribution and processes, etc..), would be dedicated for regional analysis with the support of other data and methods not presented here (e.g. Oxygen, nutrients, BGC-ARGO, Gliders, NN methods, models versus data, etc...). Here we think the selection of few figures and discussion is appropriate for potential users. Several figures were in the Supp Mat.

In short, figures 7, 8 (for global) and figure 11 (for the MedSea) present the data (here specifically when one merge different cruises). Figure 8 presents the AT and CT data along with T, S and depth (T and S properties also included in the database with their flags).
Figures 9 and 10 aimed at presenting large scale views of some derived properties (here we have selected only Revelle and Omega-Ar as example, not pH or Omega-Ca or pCO2).
Figures 12 and 13 are examples of trends in surface or at depth for few selected regions and specifically when one merges different cruises (figure 13).

This is in line with other publications of datasets in ESSD (Fassbender et al 2018; Reverdin et al, 2018 Gattuso et al 2023; Sims et al, 2023) and we keep the structure of the manuscript as submitted. Note that we did not include any comparison with methods or models as this was done in other publications, some cited in the manuscript (e.g. Lajaunie-Salla et al 2021; Chau et al, 2023 ESSD; Ulses et al, 2023 for DYFAMED time series and MedSea data; Thomas et al 2008; Keller et al, 2012; Signorini et al 2012 for SURATLANT data).

2. Please follow the same format as GLODAPv2 as much as you can. Consider making the data available in *.mat and *.nc as well. Folks with routines to import the GLODAPv2 data product should be able to adapt their routines for this new product easily.

Reply: the format of the files somehow follows the GLODAPv2, specifically for the quality flags (and also same as in SOCAT). As suggested we will add new files in Seanoe portal (and NCEI/OCADS) with .mat and .nc format for users.

3. Following GLODAP and SOCAT, please consider making this data product available through the Ocean Carbon and Acidification Data System (OCADS) at NOAA/NCEI. Doing so would enrich the product with a community-driven comprehensive metadata template, enhancing its utility and accessibility. Additionally, it would secure the benefit of a long-term archive with version control.

Reply: The data are presently secured at Seanoe and thus publicly available in a simple format. As recommended by the reviewer we will also send the files to OCADS. We have contacted NCEI/OCADS (10/10/23) and they will accept the dataset. We also indicated in the letter to the editor that once published, the files would be included in the ad-hoc GOA-ON data portal (SDG14.3.1, https://oa.iode.org/) as was previously done for data from cruises OISO, OVIDE, SURATLANT, CLIM-EPARSES.

4. Each of the individual cruise data files should also be published in a data assembly center and made publicly available.

Reply: Not sure to understand this point. As listed in the Supp Mat (Table S4), we added links of individual data files when already in a data center. Here we have synthetize the observations in only two files for easy use of these AT and CT data for the community.

5. In terms of nomenclature, please adhere to community-accepted abbreviations as outlined in Jiang et al. (2022, https://doi.org/10.3389/fmars.2021.705638). For example, use DIC instead of CT, TA instead of AT.

Reply: We prefer using the abbreviations AT and CT as recommended in the SOP (Dickson et al 2007) and used in other publications (e.g. Gattuso et al, ESSD, 2023).

Minor comments:

- Throughout the manuscript (ms): data-base --> database
Reply: Thank you, done

- Throughout the ms, pCO2 or fCO2, the p and f should be italicized.
Reply: Thank you, done

- Line 59: PgC is a unit, instead of a substance. Please add "anthropogenic carbon dioxide".
Reply: Thank you, suggestion added.

- Line 61, atmospheric CO2 is commonly reported as ppm, which is a molecular ratio, instead of a "concentration". Consider replacing it with "level" or something more appropriate. According to the IUPAC Gold Book, concentration is associated with a per-volume based unit.
Reply: Thank you, suggestion added.

- Line 65: Replace Revelle and Suess 1957, with papers that actually report these changes, e.g., DeVries, 2022; Jiang et al., 2023.
Reply: References Jiang et al 2023a added as suggested.

- Line 78: has --> have.
Reply: Corrected.

- Line 81: add Lauvset et al., 2022
Reply: References added on line 81 and in references (also commented by reviewer 2).

- Line 85: data-base --> database
Reply: Corrected.

- Line 85: add these new studies,
(a) Ma et al., (2023), ttps://doi.org/10.1029/2023GB007765;
(b) Feely, R. A., Jiang, L.-Q., Wanninkhof, R., Carter, B. R., Alin, S. R., Bednaršek, N., and Cosca, C. E. (2023). Acidification of the global surface ocean: What we have learned from observations. Oceanography, https://doi.org/10.5670/oceanog.2023.222.

Reply: Thank you, we are aware these studies. Ma et al (2023) was already in references. To our knowledge Feely et al (2023) is not yet available online (on 18/10/23) but should be soon and thus now added in references as suggested.

- Line 86: add Carter et al., 2021, https://doi.org/10.1002/lom3.10461.
Reply: Here we referred to methods for reconstructing both AT and CT. As Carter et al (2021) reconstructed only AT we did not listed this paper, but it is referenced on line 1100. Now also added on line 86 as suggested.

- Line 90: data-products --> data products
Reply: Corrected and in other lines.

- Line 107: spell out SOCOM
Reply: SOCOM was spelled out after, now moved before the acronym.

- Line 116: add Ma et al. (2023) and Feely et al. (2023). For more details, see above.
Reply: References added

- Line 122: 2013b --> 2023b
Reply: Thank you, corrected

- Replace data citations with citations to the paper. That will allow readers to access more information of the product.

For example:

Jiang, L.Q., Feely, R. A., Wanninkhof, R., et al.: Coastal Ocean Data Analysis Product in North America (CODAP-NA, Version 2021) (NCEI Accession 0219960). [indicate subset used]. NOAA National Centers for Environmental Information. Dataset. https://doi.org/10.25921/531n-c230. Accessed [date]. 2020.

Should be replaced with:

Jiang, L.-Q., Feely, R. A., Wanninkhof, R., Greeley, D., Barbero, L., Alin, S., Carter, B. R., Pierrot, D., Featherstone, C., Hooper, J., Melrose, C., Monacci, N., Sharp, J. D., Shellito, S., Xu, Y.-Y., Kozyr, A., Byrne, R. H., Cai, W.-J., Cross, J., Johnson, G. C., Hales, B., Langdon, C., Mathis, J., Salisbury, J., and Townsend, D. W. (2021). Coastal Ocean Data Analysis Product in North America (CODAP-NA) – an internally consistent data product for discrete inorganic carbon, oxygen, and nutrients on the North American ocean margins. Earth System Science Data, 13(6), 2777–2799. https://doi.org/10.5194/essd-13-2777-2021.

Reply: Thank you, corrected and on line 1113.

Reference in this review not listed in the MS:

Thomas, H., A.E.F. Prowe, I.D. Lima, S.C. Doney, R. Wanninkhof, R.J. Greatbatch, U. Schuster and A. Corbière, 2008. Changes in the North Atlantic Oscillation influence CO2 uptake in the North Atlantic over the past two decades, Global Biogeochem. Cycles, 22, GB4027, doi:10.1029/2007GB003167

Signorini, S. R. , S. Häkkinen, K. Gudmundsson, A. Olsen, A. M. Omar, J. Olafsson, G. Reverdin, S. A. Henson, C. R. McClain and D. L. Worthen, 2012. The Role of Phytoplankton Dynamics in the Seasonal and Interannual Variability of Carbon in the Subpolar North Atlantic:¬ A Modeling Study. Geosci. Model Dev., 5, 683-707, doi:10.5194/gmd-5-683-2012

Lajaunie-Salla, K., Diaz, F., Wimart-Rousseau, C., Wagener, T., Lefèvre, D., Yohia, C., Xueref-Remy, I., Nathan, B., Armengaud, A., and Pinazo, C.: Implementation and assessment of a carbonate system model (Eco3M-CarbOx v1.1) in a highly dynamic Mediterranean coastal site (Bay of Marseille, France), Geosci. Model Dev., 14, 295–321, https://doi.org/10.5194/gmd-14-295-2021, 2021.

Keller K., F. Joos, C. Raible, V. Cocco, T. Frolicher, J. Dunne, M. Gehlen, L. Bopp, J. Orr, J. Tjiputra, C. Heinze, J. Segscheider, T. Roy and N. Metzl, 2012. Variability of the Ocean Carbon Cycle in Response to the North Atlantic Oscillation. Tellus B, 64, 18738, doi:10.3402/tellusb.v64i0.18738.

---

## Author Comment (AC2)

Response to Reviewers' comments on the manuscript:

A synthesis of SNAPO-CO2 ocean total alkalinity and total dissolved inorganic carbon measurements from 1993 to 2022, Nicolas Metzl et al., MS No.: essd-2023-308, MS type: Data description paper

Reply to Reviewer 2, Marta Alvarez (in purple from reviewer, in black our reply)

Reviewer 2 (Marta Alvarez): posted 1/10/23:

Dear authors,

this is a massive effort that deserves publication, the SNAPO CO2 French service is pivotal to the oceanographic community in France, it merits sustained funding and recognision. It is very remarkable that most of the gathered data is already published both in papers and/or in a public database.

Most of my comments in the annotated pdf relate to organization issues. The text can be organized in a slight different way to highlight for example the quality of the measurements and provide a little bit of more (or less) information in some sections.

So please take the "major revision" as an improvement to highlight the SNAPO good practices.

An issue the authors would consider is the name of the database and its corresponding updates... if the current datebase is name SNAPO-CO2, which would be the next update name? .... so authors may consider SNAPOv1 or SNAPO 1993-2022 or another option.

I also think that the name of the database should not be the same as the name of the service ... but this is up to the authors and their strategic plan to sustain and highlight the SNAPO performance and reliability.

Best regards

We thank Marta Alvarez for her supportive review and suggestions.

Below we list her comments (in purple), suggestions and questions addressed in the pdf document (identified for each line) and our reply (in black).

;;;;;;;;; On the pdf document

Line 1: maybe add "French SNAPO-CO2 service"... it would focus and strengthen the need and continuity for this service

Reply: Thanks for the suggestion. However, there are some cruises conducted with support of International (e.g. AWIPEV) and we prefer to not specify this is "French". In the future, we would probably add data from international cruises as well.

Line 41: In my head Global change includes climate change ... in any case .. I would also point to "find carbon based solutions or mitigation procedures" .. something about that
Reply: Climate change is generally for warming and global change include warming, change in circulation, acidification, productivity, species, etc…. As suggested we use only "Global Change". We also add the MCDR issue (good idea).

Line 42: along with basic ancillary data (time and space location, pressure temp and salinity)
Reply: Thanks, words added.

Line 42: I think this information in the abstract is too much ... I would stress that the database considers only discrete analysis with samples from
- water column
- surface underway
I guess that if you mention one program as OISO o CLIM-EPARSES the others could be a bit dissappointed.
Reply: Thanks for the suggestion. Deleted in the abstract as this is clear in the MS.

Line 49: This is a bit weird .. most of the repositories assign a DOI to each data set, not to a sort of cathalog ... it might be good to clarify a little bit this point ...
In addition, this is a collection from 1993 til 2022, are you planning to update regularly the SNAPO collection ? .. accompanied with another ESSD paper?
I know there are many issues with releasing the data publicly .. but it might be good for the SNAPO future govermental support, to slightly suggest that the SNAPO-CO2 collection will be updated some how.
Reply:  Yes, we are planning to update this dataset, as indicated in the conclusion (lines 1171-1172 original MS). Not sure we need to specify this in the abstract. As suggested we clarify the information in the abstract concerning the dataset repository. The DOI and reference listed in the abstract was suggested by the editor once the paper was submitted.

Line 53: BG or BGC Argo floats?
Reply: Thank you, corrected "BGC-Argo" in the abstract and the MS

Line 61: I would say "global change" .. it comprises everything
Reply: Changed to.. "change".

Line 68: by the World Meteorological Organization ...
Reply: Corrected

Line 77: the future evolution of global change?
Reply: Rephrased

Line 81: the 2022 update is already published and released
Reply: Reference added (also noticed by reviewer 1).

Line 85: publicly available consistent and quality controlled data bases
I would take out GLODAP , as it is already mentioned
there are also other efforts as
- CODAP-NA
Jiang, L.-Q., R. A. Feely, R. Wanninkhof, D. Greeley, L. Barbero, S. Alin, B. R. Carter, D. Pierrot, C. Featherstone, J. Hooper, C. Melrose, N. Monacci, J. Sharp, S. Shellito, Y.-Y. Xu, A. Kozyr, R. H. Byrne, W.-J., Cai, J. Cross, G. C. Johnson, B. Hales, C. Langdon, J. Mathis, J. Salisbury, and D. Townsend. 2021. Coastal Ocean Data Analysis Product in North America (CODAP-NA) - An internally consistent data product for discrete inorganic carbon, oxygen, and nutrients on the U.S. North American ocean margins, Earth Syst. Sci. Data, 13, 2777-2799, https://doi.org/10.5194/essd-13-2777-2021.

- CARIMED
unfortunately unpublished!

Reply: Revised as suggested. The CODAP-NA reference was listed on line 1113 and 1544: new reference to ESSD now added (also suggested by reviewer 1). For CARIMED, in progress, we thought it would be useful to inform the community, but as suggested we delete this reference.

Line 93-97: with this and the previous paragraphs I get a bit confused ... I know the SNAPO data could be used to reproduce surface or interior data ... but I think it could be better for the ms to comment on the likely uses of the SNAPO data base instead of broadly commenting the benefits of GLODAP or SOCAT ... first shortly comment on GLODAP and SOCAT or others .. and then give a brief description about the SNAPO corresponding contribution to products / processes .. etc ...

Reply: These paragraphs aimed at recalling the support of quality control data in SOCAT and GLODAP for various reconstructions (using SOCAT or GLODAP independently). On line 93-97 we also inform that SOCAT and GLODAP have been successfully coupled to constraint new methods for reconstructing pH, Omega, etc... We would like to keep this information that was not commented by reviewer 1.

Lines 117-126: As commented before.. the previous paragraphs give too much information about GLODAP and SOCAT
this paragraph here is the main main one ... why SNAPO-CO2? how can it contribute to other international efforts
One important thing is .. will SNAPO-CO2 be uploaded to IODE SDG 14.3.1 ..? I know they are only interested in surface pH data ... but they accept calculated pH!

Reply: After describing SOCAT and GLODAP, this paragraph explains shortly why we propose to synthetize AT-CT data in order to complement observations NOT in SOCAT or GLODAP (and motivate community to add surface AT-CT data in SOCAT, not only fCO2, but this clearly needs support at international level as recently recalled by Bakker et al 2023). Concerning SDG 14.3.1 this is correct. Once the paper published, the data will be sent to the SDG portal (https://oa.iode.org/) as was done for several cruises. Note that, when we submitted the article, in our letter to the editor last 1st August, we mentioned: "The aim of this work is to describe the data assemblage, the data quality control and to discuss some potential uses of this dataset. Once published, the files would be also included in the ad-hoc GOA-ON data portal (SDG14.3.1).".  In addition, reviewer 1 suggests send the data to NCEI/OCADS and this will be also prepared. Therefore, the data would be available on several platforms for users: Seanoe, GOA-ON and OCADS.

Lines 129: the same instrument all over those years .. is a robust one!!!
could be worthy to mention here again the quality control . use of CRM

Reply: Yes we used the same instruments (several versions since the 90s used in the laboratory and on-board based on the same technic and laboratory made closed-cell). The quality control and the use of CRM are described in dedicated sections (i.e. not recall again here, also indicated in the abstract). FYI, concerning the instruments, as there is more and more samples to analyze we have now (since September 2023) two instruments working in the lab.

Line 131: The terms weather and climate goals for CO2 variables were first used by
Newton, J., Feely, R., Jewett, E., Williamson, P., Mathis, J., 2015. Global Ocean Acidification Observing Network: Requirements and Governance Plan.
Reply: Thank you, reference added line 131 (as was listed on line 375 and 576).

Line 134: I suggest this section is merged with the next one
only one section
Data and methods

- cruise overview
- sampling
- analytical procedure
- quality control
    + accuracy
    + reproducitility and precision
    + inter comparisions
    + quality flags assigned
I suggest the section about pCO2 calculation should be out of the "Data and methods"

Reply: This is a possibility, not mentioned by reviewer 1. However this would not change the structure and information in each section. We'd like to keep as submitted. The section "data collection" aimed at presenting shortly all cruises in a single table (table 1) and a map (figure 1). A reader would easily see where the data are located if interested in only one specific region/period. The section includes information on the way samples were collected. This explains the short title of this section: "data collection". At the end we specify that some data were obtained with different techniques.

Line 137: please refer to Table S1
I think Table 1 in the manuscript and Table S1 in the supplement are the same
I think Table S1 & S2 can be merged and showed in the manuscript ... but probalbly in horizontal mode.
Reply: Added Table S1 on line 137. Table 1 is a synthesis of cruises/projects whereas Table S1 and S2 were dedicated to present in more detail the period of each cruise (e.g. PIRATA-FR), PIs of cruises and CRM used for each cruise/project. We don't think it is relevant to include this information in the main text.

Lines 144-147: samples not included in this manuscript should not be described ...
I suggest a general introduction to the SNAPO service in the introduction, why is was launched, when, who runs it?... and which is the specific mission => support to measure CO2 discrete samples from cruises or experiments .. and then give some examples and references as the ones for the experiments mentioned in this paragraph
Reply: We understand this comment as we indicate here that some measurements were performed for other projects but data not include in the dataset (for the reason explained).  Here we mentioned few other projects (culture or mesocosm experiments) but we do not describe these projects in detail (only 3 lines and few references). A reader may be interested in the data (e.g. DUNE project) or for asking for new analysis in future (e.g. for MCDR project ?). This was not commented by reviewer 1 and we'd like to keep this information. On the other hand the aim of this paper is to describe the SNAPO-CO2 dataset (not the SNAPO-CO2 service and why it has been developed in our laboratory).

Line 150: I think this is not relevant for the description of the data base and could be removed
Reply: We think it is relevant to inform that new instruments were developed to measure AT underway in surface water on SOOP (somehow like pCO2). This is only one sentence and might motivate for a synthesis of AT-CT data underway in the future, i.e. the same way as done for underway fCO2 in SOCAT.

Lines 158-160: I do not understand ... those cruises were measured at sea?
this issues should be mentioned in the sampling section of the general methods section ...
Reply: At the end of this section we inform that data in this synthesis were obtained using the same technic except for few cruises (e.g. PENZE or DYFAMED in 1998-20023 or SURATLANT in 1992-1996) as specified in dedicated sections and we decided to include these data in the synthesis. Here, we also specified that for OISO and CLIM-EPARSES cruises the data were measured underway at sea. We

think this is part of the "Data collection" section as we refer that underway surface measurements were also obtained (line 149) and detail of sampling at sea for these cruises can be found in publications listed in Table S3 (thus probably not relevant to recall again).

Line 162: this is the same Table as S1 ... as commented .. S1 and S2 could be merged and shown in the main ms
Reply: as explained above, Table 1 is a synthesis of cruises/projects whereas Table S1 and S2 give more details of periods, PI, or CRM used. We prefer to list in the main text the most synthetic table (Table 1).

Line 190: defined? in a row above you use North Atl ... please use the same acronyms
Reply: Thank you, North Atl corrected.

Line 228 (figure 1): could be nice to mention the name of the data base ... SNAPO-CO2
please locate with a square in the global map the location of the inset
Reply: Good idea thank you: SNAPO-CO2-V1 added in the legend. White square of the insert map added on the global map (figure 1 revised).

Line 235-240: please first provide the genearl setting and then the exceptions
the SNAPO service should be commented in the intro
within the "analytical procedure" subsection more info about the potentiomatric titration could be given ... and then a comment on wether the samples were measured on board or in Paris ... maybe this info could be given in one of the Tables.. the big Table 1.
Reply: The section has been modified to provide first the information for measurements at SNAPO-CO2 service since 2003 and the underway measurements at sea. We must indicate that some data, prior to year 2000 have been added in the dataset to complement 2 time series (only for DYFAMED and SURATLANT). As suggested, we now specify on board measurements in table 1 (Good idea): a * is added in Table 1 to inform on data measured at sea (note in the legend)

Line 240: the method itself merits a longer description with pros and cons compared to the more commonly used  is important to describe the control of the accuracy and drift for every batch of analysis
Reply: Not sure to understand what we should add for the description, the comparisons or the drift for batches already documented in the manuscript.  The method (used for discrete sample or onboard) has been described in previous publication (e.g. Goyet el al, 1991; Metzl et al, 1991; Metzl et al, 2006; Corbière et al., 2007; Kapsenberg et al 2017; Reverdin et al, 2018; etc…).

Concerning the comparison with other methods, it has been first intercompared (12 laboratories) in the frame of the JGOFS-IOC Advisory Panel on Ocean CO2 (Poisson et al 1990; UNESCO 1990), as outline line 243 in this section.  In the UNESCO report it is mentioned that "It is noticeable that the scatter of mean data for potentiometric titration is no greater than that of extraction techniques (gas chromatography, manometry, coulometry)". More recently the SNAPO-CO2 method has been also compared with other methods during an international intercomparison conducted in April 2017; this was noticed by Reverdin et al (ESSD 2018): "A recent international intercomparison on two shared water batches (spring 2017) suggests that the LOCEAN analysis presents a small negative bias both for At and DIC (Emily Bockmon and Andrew Dickson, personal communication 2018), but not in a very similar range of values to the ones observed during SURATLANT." Unfortunately, at present, we cannot refer to an official publication of this 2017 intercomparison (as was done by Bockmon and Dickson, 2015) but we expect this might be published for the next SNAPO-CO2-v2 version.

On the other hand, there is a specific section "3.3 Inter-comparison", where we describe several comparison with other methods, especially for the North Atlantic (SURATLANT or OVIDE cruises see

Table 3), recalling that our measurements were performed in the laboratory whereas those from the IIM/Vigo group were done on-board. Finally, the comparison of calculated fCO2 with fCO2 measurements (section 3.5) provides also a kind of inter-comparison of independent data.

Concerning the "drift for every batch of analysis", we think figure 2 informs that there is no drift, but obviously some noise. For this reply we have prepared a new figure (Figure R1 below, not in the MS) showing the results of 136 analysis for Batch 182 (one used for more than a year from 28 May 2019 to 20 July 2020).

[Figure]

Figure R1: Results of all measurements for Batch 182 (CRM reference values are: CT= 2039.1 µmol/kg; AT = 2230.9 µmol/kg). For 136 analysis of the same Batch, the mean values for CT is 2039.08 (±3.6) µmol/kg and for AT = 2230.83 (±3.6) µmol/kg. The red line indicates the "drift": less than 0.2 µmol/kg/yr for both AT and CT.

Another way to check the drift is presented below in Figure R2. Here we have selected all Batches used for the period 2005 to 2023.

[Figure]

Figure R2: Results of all measurements for Batch 66 to 197. Each point is the difference between measured and reference value. For 1150 analysis the mean differences are: CTmes-CTref= +0.11 (±3.48) and ATmes-ATref =-0.039 (±3.21) µmol/kg. Color code is Batch number.

Line 245: Sorry, I am nor famiiar with the potentiometric technique … CRM should not be used as a calibration point, but rather as a checking point .. I am very aware that many labs, as there is no other option, proceed that way ..
Does your procedure mean that if the CRM analysis is far from the certified value .... the samples values are corrected accordingly?
Please add a reference by Dickson about the reliability of CRMs bla bla
Reply: When a CRM measurement is far from the certified value (example are shown in figure S2), this indicates a problem with the analysis (e.g. a very small, almost undetectable bubble in acid delivery that leads larger difference for both AT and CT, up to 8-15 μmol/kg). Therefore, such CRM measurement is no longer used as indicated on line 251. As suggested, a reference has been added for reliability of CRMs following: "The concentrations of CRMs we used vary between 2193 and 2426 μmol kg$^{-1}$ for AT and between 1968 and 2115 μmol kg$^{-1}$ for CT corresponding to the range of concentrations observed in open ocean water. The CRMs accuracy, as indicated in the certificate for each Batch, is around ±0.5 μmol kg$^{-1}$ for both AT and CT (www.nodc.noaa.gov/ocads/oceans/Dickson_CRM/batches.html). "

Line 248: it is a bit confusing paragraph, first you talk about 724 analysis and then about 985 analysis I would suggest first describe the whole analysis, I think they are in Figure S2 and then after a brief justification .. Figure 2 ...
in this sense why do you prefer to show Fig2 instead of Fig S2?
Reply: Thank you for this comment. Figure 2 was presented as an example of CRM analysis (for the main text), i.e. a selection of 724 analysis for 2013-2020 whereas Figure S2 shown all results for 2005-2020. Recently, we have added data in 2021 and 2022 and the CRM measured were listed in Table S2 (Batch 191, 196, 197) but were not included in Figure S2.  This is now corrected for Figure S2. Concerning Figure 2 we think it is appropriate to show only part of the CRM results (for clarity of the message). When showing all results the figure is not so clear because of the concentration ranges (see Figure R3 blow). We prefer to show only a figure with CRM concentration in the same range. This is now shown in a new figure 2 for 2013-2023 with CRM measured in 2021-2023 included (see figure R4 that is now Figure 2 in the revised paper). We have corrected the legend and statistics.

[Figure]

Figure R3: Same as Figure 2 in the main MS but for all CRM in 2005-2023.

[Figure]

Figure R4 (now new figure 2): **Figure 2:** $A_T$ (a) and $C_T$ (b) analyses for different CRM Batches measured in 2013-2023. For these 965 analyses the mean and standard-deviations of the differences with the CRM reference were -0.1 (± 3.4) µmol kg$^{-1}$ for $A_T$ and 0.1 (± 3.7) µmol kg$^{-1}$ for $C_T$.

Line 249: the STD is important but also the deviations from the certified value ... the mean is around cero ?
Reply: Yes, the mean is around 0 as indicated in legend of figure 2.

Line 253: which CRM analysis ?
Reply: this referred to all CRM measurements not specific ones. Corrected.

Line 255: I do not understand this phrase .. you mean that the batch of analysis could be as long as 2 to 7 days? ... this is something that could be described in the analytical procedure
Reply: Depending the number of samples for each cruise it takes between 2 to 15 days for the analyses. The standard deviation of all CRM  is around 3.5 µmol kg-1 (as indicated in this section), but for some cruises we obtained better result, less than 3 µmol kg-1, as published in several paper (reference listed on line 256-257). Sentence revised.

Line 256: There are too many numbers in these paragraphs which could be a bit confusing
I suggest providing a general number .. mean and std for the whole CRM analysis based in Figure S2 I guess ... and then provide specific examples maybe using Figure 2
I do not think is useful to link the info in Fig 2 with cruises or papers
maybe just saying that the specific info about the SNAPO results are provided in the publication regarding the cruise given in Table 1
Reply: The mean std for the whole CRM was given at start in this section. There are only 3 numbers listed in this section (not so much). For clarity we have deleted the number listed at the end. You are right, readers can find this information in the associated publication.

Line 278-280: Suggestion .. provide as text on the figure .. the mean and std for each batch.. only one significant number not two

Reply: We think that adding numbers (mean and std) for each batch in figure 2 would add a lot of information (maybe too much on the figures, data are now presented for 14 Batches). Significant numbers have been corrected (now only one), also done in other parts of the paper (Table 6).

Reply: Thank you for this comment. "Reproducibility" referred to measurements on same sample by different instruments and/or different groups (e.g. UNESCO 1990; Bockmon and Dickson, 2015). "Repeatability" referred to results from replicates samples analyzed on the same instrument, here SNAPO-CO2 (e.g., Kapsenberg et al., 2017); this includes all uncertainties related to the sampling, transport duration of samples, storage, and analysis. In this section most of the information present results on repeatability. We have listed few results of reproducibility (in the inter-comparison section). For clarity we have changed the title of section 3.2 and the legend in Table 2.

Reply: Yes in fact the section describes only Repeatability. Corrected.

Reply: Thank you there were repeated references in the text, table 2 and figure 3. Section revised, references deleted as suggested.

Reply: This is correct; we indicate accuracy based on CRM of ±3.5 µmol kg-1 in the previous section. Here we somehow resume that based on CRM and replicates (that include error associated to sampling) we estimate data in this synthesis to be consistent better than 4 µmol kg-1. Interestingly, this is the same as in GLODAP (Lauvset et al, 2022). Sentence has been revised.

Reply: Thank you, corrected in Table (legend).

Reply: STD informs on the noise from "duplicates" that include sampling, storage, analysis, transport, etc… and it is the same information for each cruise/project. The mean of the differences would be somehow different when based on comparing 12 samples taken at same depth (for OUTPACE, SOMBA), duplicate for each period (for times series) or duplicate/triplicate/…/sixplicate from underway surface measurements at same location (OISO, CLIM-EPARSES).

Reply: Moved to a note.

Reply: Thank you, deleted.

Reply: Thank you, deleted.

Reply: Thank you, revised.

Line 371: There several ways of performing intercomparisions .. in your case all the casestudies correspond to sort of double check with other techniques
I suggest a short intro for this section and then present the different cases as
A) B) C)
In my opinion intercomparisions do tell whether measurements are climate or weather goal quality
... they say if lab-to-lab or technique to technique results agree to within a given numbers ...
I would suggest to take out from this section any reference
Reply: There was a short introduction in this section to recall why it is important to conduct inter-comparisons for AT-CT (and when available with fCO2 or pH calculated from AT-CT). Here we show some examples from our dataset. One sentence has been deleted and references already mentioned at the end of the introduction deleted. As suggested we add sub-sections for each comparison (CHANNEL, SURATLANT, OVIDE and PENZE).

Line 371: I wouls suggest biases or systematic differences
Reply: Agree, sentence revised.

Line 372: Delete
Reply: Deleted

Line 383: suggestion present the information regrding the intercomparision with the same order .. project , techniques used, CRMs used yes or no?, differences with SNAPO data, mean and STD
Reply: This is somehow what is presented for CHANNEL or SURATLANT. Not changed.

Line 386-387: delete
Reply: We'd like to keep this sentence: this was a short conclusion for CHANNEL inter-comparison indicating that long-term estimate could be evaluated using CHANNEL data with other data when available if a user is interested to conduct such analysis (e.g. extend the work from Kitidis et al 2019).

Line 390: coulometry is for CT  what technique was used for AT?
Reply: For AT potentiometric method was used by other groups (Table A1 in Reverdin et al, 2018). Not specify here because we only informed on different technics (for CT).

Line 392: STD?
Reply: Yes this is STD, now specify

Line 393: delete ... if any publication as Reverdin .. presents the comparision or as Marrec above ... just add the comment at the end of the corrsponding paragraph.... so many repeats of the same paper reference in the same paragraph is a bit confusing
Reply: Thank you, deleted.

Line 397: is this acronym used in other parts
Reply:  Yes, acronym used in the text and tables

Line 397: According to rr et al 2018 .. the total uncertainty in calculated CT from pH (0.01 uncert) and TA  (2 uncertainty) is 0.6%
therefore for about 2300 umol/kg CT .. it would be 13.8 umol/kg of uncertainty ... this is a maximum .. but it is realistic .. so I would take out any comparison with calculated data
if you think keeping the CT comparison is useful ... please give info about the CO2 constants used.
Reply:  Thank you for this comment. As mentioned, for OVIDE, we compared the measured CT with calculated CT values from AT/pH pairs (here we used quality controlled data from GLODAP). We have

also checked the error for CT calculation based on Orr et al (2018). For example, results for one sample (station 34 OVIDE 2008) and using Lueker et al (2000) constants:

| | Salinity (PSU) | Temp (C) | Depth (m) | PHO (µmol/kg) | SIL | AT (µmol/kg) | pH (TS) |
|---|---|---|---|---|---|---|---|
| Value | 35.752 | 16.623 | 7 | 0.04 | 1.164 | 2352.43 | 8.144 |
| Error | 0.01 | 0.01 | | 0.01 | 0.01 | 2 | 0.01 |

This leads to CT calculated of 2059.34 µmol/kg with an error of 8.78 µmol/kg. The CT calculated in the GLODAP dataset is 2059.3 µmol/kg for this sample, i.e. same as our calculation.
As suggested, we have added the error information in this section and added a reference (Orr et al, 2018); for other details on OVIDE data and calculations readers could be refer to GLODAP or the publications we have listed for OVIDE (Pérez et al., 2010, 2013, 2018; Vazquez-Rodriguez et al., 2012). Based on the differences for several cruises (listed in the Table 3), we conclude that, given the uncertainty in CT calculations, the data from SURATLANT and OVIDE in the SNAPO-CO2-v1 dataset could be merged with other data (e.g. GLODAP data) to analyze the seasonal variations or long term changes in the NASPG. As an example we show that when associated for year 2010, the data inform on a rapid change of CT and fCO2 (Figure 4) associated to a strong biological event occurring that year. For K1, K2 we used constants from Lueker et al (2000) and we have added this reference as suggested.

Line 423: sometimes the authors use pCO2 and others fCO2 .. do you make any difference in the calculations?
please homogeneize
Reply:  Thank you; we have change to fCO2 here as we refer to SOCAT. This has been corrected in other sections. For most results we calculate fCO2 (e.g. Figure 4). We used pCO2 only when the original data were listed as pCO2 (e.g. comparison in figure 6).

Line 479: please clarify in the corresponding C) section if the LOCEAN samples correspond to deep surface or radom .. .is not clear in the text
just one significant figure after the dot would be enough
Reply:  For OVIDE cruises most of the LOCEAN samples were taken in surface (to complement the SURATLANT time series in summer), but we also sampled deep waters at stations (as indicated in the introduction of this section) especially for inter-comparison with previous cruises (same station around 60°N) and data from IIM/Vigo group. The OVIDE data are thus for both surface and water column (as indicated in Table 1). Values in Table 3 revised (only one decimal)

Line 487:I guess it could be more scientiifc if both regression lines are presented as
y+/-error = intercept (+/-error)  + slope (+/-error)  R^2 and p value
Reply: The PENZE data in 2019-2020 are from samples not measured at SNAPO-CO2. Before including these data in the synthesis we checked if it was coherent with the SNAPO-CO2 data. Note that for 2019-2020 no CT is available (flag 9 in the data files), but the AT data may be important for further analysis. As suggested, the regressions are (and added in manuscript):
In 2011: N samples =78: AT = 51.525 (±0.944) * S + 583.95 (±19.94) (r2= 0,975)
In 2019: N samples= 70: AT = 54.022 (±1.018)* S + 450.23 (±31.53) (r2 =0,976)
We have also slightly revised figure 5 (Label SBR data in 2019-2020, not only 2019).

Line 509:Quality control assigned flags
I think this is a more proper title as previous sections dealt with quality control procedures ... the flags are the final step
Reply: Good suggestions, revised accordingly.

Line 512:please provide here the flag definitions 2-OK 3- questionable 4- bad  9 mnot measure ..
Reply: Definitions added (it was also indicated later in the section).

Line 606: I suggest this section to be moved after the spatial and temporal description of the data set
I think is related with methods ... but could stand alone
Reply: In this section we also compared calculated properties (here fCO2 and pH) with independent
data; we prefer to keep this section 3.5 after 3.3 and 3.4 once quality flags have been fixed
(calculations for fCO2 and pH performed with flag 2 data).

Line 614: state that this is the CHANNEL project to link with figure 6
Reply: CHANNEL project added

Line 619: please comment on the CO2 constants ... and I think also based in Orr et al 2018 .. the
uncertainty in calculated pCO2 from CT and AT Table 5 is 3.5% .. so pCO2 calcualted and measured
are coherent
Reply: The main aim of the comparison of CHANNEL data described and presented in figure 6 was to
show that there is no shift or drift of the differences between measured and calculated pCO2 at
seasonal scale and from year to year. Details on constants used were described by Marrec et al
(2014) and we think it is not relevant to recall this again. For your information, Marrec et al (2014)
used constants from Mehrbach et al. (1973) refitted by Dickson and Millero (1987). The error
associated to the pCO2/fCO2 calculations using AT/CT pairs (with AT and CT error of 3 µmol/kg based
on SNAP CRM) is 13 µatm (Orr et al, 2018). Your conclusion is correct, given this uncertainty, the
results for measured and calculated pCO2 are coherent.

Line 657-659: give info about  - CO2 constants - mean difference and std
Reply: Here we first recalled the analysis by Merlivat et al (2018) based on 2013-2015 CARIOCA data;
then we presented new results extended for recent years (2013-2018) using all AT-CT data in the
SNAPO-CO2-v1 dataset. Merlivat et al (2018) indicated a standard deviation difference of 4.4 µatm
using the constants from Mehrbach et al refitted by Dickson and Millero (1987) "and as
recommended by Álvarez et al. (2014) for the Mediterranean Sea". Again, we think it is not relevant
to recall the detail of the constants here. However, as suggested, we now have specified that for new
calculation presented here for 2013-2018 (Figure S4) and for coherence with other fCO2 calculations
(figure 4 for NASPG) we used the constants from Lueker et al (2000). This reference is now added.
For your information, we have compared the results with constants from Mehrbach et al refitted by
Dickson and Millero (1987) (MDM) and we obtained the same results (Figure R5 below):

Mean (and std) differences for 67 co-located samples (fCO2cal-fCO2CARIOCA):
    fCO2 calculated with Lueker          fCO2 calculated with MDM
    -3.72 µatm (±10.76)                   -3.28 µatm (±10.78)

[Figure]

Figure R5 (same as Figure S4): Measured fCO2 versus calculated fCO2 for 67 co-located samples: Black dots and dashed line: using Lueker et al (2000); Red triangles and red line using Mehrbach et al refitted by Dickson and Millero (1987).

Line 669: Plese separete the fCO2 cases from the pH cases .. it could be nice to have two separate subsections
Reply: As this was specific for the MedSea and the DYFAMED/BOUSSOLE location, we prefer to keep this short sentence at the end of this section.

Line 669: this is very nice but out of the scope of this section .. it can be (and I think already is) in the intro section about the SNAPO general objectives
Reply: Here we mentioned few studies using AT-CT data for interpreting response of marine species or corals to acidification. As the TARA-Pacific data are included in this synthesis we think useful to refer original publications (not listed in the introduction). Not changed.

Line 714: maybe :Global distribution and relationships fro At and CT based on the SNAPO CO2 database
Reply: Thanks for the suggestion. Section title changed: Global distribution and relationships from AT and CT based on the SNAPO-CO2 dataset

Line 878: pleae state somewhere that Omega is calculated at in situ conditions of Temp Sal and pres
Reply: This is now added when introducing Omega.

Line 976: please use the same acronyms for the areas here and in Figure 12
Reply: Corrected: We added acronyms in Table 6 (same as for figure 12). We changed also CT trend number in Table 6 (only 2 decimals)

Line 1035: already defined?
Reply: It was not defined, now added, thank you.

Line 1050: could be deleted
Reply: we wanted to inform the community on ongoing projects. CARIMED deleted as suggested and GLODAP added.

Line 1075: a better tytle promoting SNAPO-CO2 could be great!
SNAPO-CO2 data use to validate BGC data from autonomous instruments.
Reply: Thanks for the suggestion but we prefer to keep the original title as the AT-CT data could be used to compare and validate any autonomous instruments not only BGC-ARGO (e.g. mooring such as CARIOCA, Gliders, Saildrone, etc…). Here we show an example (figure 14) as these samples were dedicated to compare with the BGC-ARGO launched during the ACE cruise.

Line 1078: BGC?
Reply: Correct. "BGC-Argo" corrected in the manuscript.

Line 1078: between the period
Reply: Corrected.

Line 1111: delete

Reply: Deleted as suggested.

Line 1166: this comment may go in the next section

Reply: The next section "Data availability" should only inform the link to the data. No change in the conclusion.

Reference added in this reply and in the manuscript:

Orr, J. C., Epitalon, J.-M., Dickson, A. G., and Gattuso, J.–P.: Routine uncertainty propagation for the marine carbon dioxide system, Marine Chemistry, Vol. 207, 84-107, doi:10.1016/j.marchem.2018.10.006., 2018.

Lueker, T. J., Dickson, A. G., and Keeling, C. D.: Ocean pCO(2) calculated from dissolved inorganic carbon, alkalinity, and equations for K-1 and K-2: validation based on laboratory measurements of CO2 in gas and seawater at equilibrium. Marine Chemistry 70, 105-119. https://doi.org/10.1016/S0304-4203(00)00022-0, 2000.

Reference added in this reply not in the manuscript:

Bakker, D., R. Sanders, A. Collins, M. DeGrandpre, T. Gkritzalis, S. Ibánhez, S. Jones, S. Lauvset, N. Metzl, K. O'Brien, A. Olsen, U. Schuster, T. Steinhoff, M. Telszewski, B. Tilbrook, D. Wallace, 2023. Case for SOCAT as an integral part of the value chain advising UNFCCC on ocean CO2 uptake http://www.ioccp.org/images/Gnews/2023_A_Case_for_SOCAT.pdf

Metzl, N., C. Beauverger, C. Brunet, C. Goyet and A. Poisson, 1991. Surface water pCO2 in the Southwest Indian Sector of the Southern Ocean: a highly variable CO2 source/sink region in summer. Marine Chemistry, 35, 1-4, 85-122. https://doi.org/10.1016/S0304-4203(09)90010-X